# The Future is Log-Gaussian: ResNets and Their Infinite-Depth-and-Width Limit at Initialization

**Mufan (Bill) Li**[*]
University of Toronto,
Vector Institute

**Mihai Nica**[*]
University of Guelph,
Vector Institute

**Daniel M. Roy**
University of Toronto,
Vector Institute

## Abstract

Theoretical results show that neural networks can be approximated by Gaussian processes in the infinite-width limit. However, for fully connected networks, it has been previously shown that for any fixed network width, $n$, the Gaussian approximation gets worse as the network depth, $d$, increases. Given that modern networks are deep, this raises the question of how well modern architectures, like ResNets, are captured by the infinite-width limit. To provide a better approximation, we study ReLU ResNets in the infinite-depth-and-width limit, where *both* depth and width tend to infinity as their ratio, $d/n$, remains constant. In contrast to the Gaussian infinite-width limit, we show theoretically that the network exhibits log-Gaussian behaviour at initialization in the infinite-depth-and-width limit, with parameters depending on the ratio $d/n$. Using Monte Carlo simulations, we demonstrate that even basic properties of standard ResNet architectures are poorly captured by the Gaussian limit, but remarkably well captured by our log-Gaussian limit. Moreover, our analysis reveals that ReLU ResNets at initialization are hypoactivated: fewer than half of the ReLUs are activated. Additionally, we calculate the interlayer correlations, which have the effect of exponentially increasing the variance of the network output. Based on our analysis, we introduce *Balanced ResNets*, a simple architecture modification, which eliminates hypoactivation and interlayer correlations and is more amenable to theoretical analysis.

## 1 Introduction

The characterization of infinite-width dynamics of gradient descent (GD) in terms of the so-called Neural Tangent Kernel (NTK) [1–10] represented a major breakthrough in our understanding of deep learning in the large-width regime. Before the identification of infinite-width limits, the theoretical study of deep learning had long been hindered by the apparent analytical intractability of gradient descent and variants acting on the nonconvex objectives used to train neural networks. Despite this progress, evidence suggests that deep neural networks can outperform their infinite-width limits in practice [11], particularly when the depth of the network is large. These observations motivate the study of other approximations that may close the gap.

Several alternative limits have been proposed. Around the time of the discovery of the NTK limit, mean-field limits were also characterized [12–15], and more recently have been linked with the NTK limit [16]. Yang and Hu [17] describe a family of infinite-width limits indexed by the scaling limits of initial weight variance, weight rescaling, and learning rates. This family includes both the NTK and mean field limits. One motivation for studying these alternative limits is that they yield a notion of feature learning, which provably does not occur in the NTK limit [17].

---

[*]Equal contribution authors.
Correspondence: `mufan.li@mail.utoronto.ca`; `nicam@uoguelph.ca`; `daniel.roy@utoronto.ca`.

35th Conference on Neural Information Processing Systems (NeurIPS 2021).

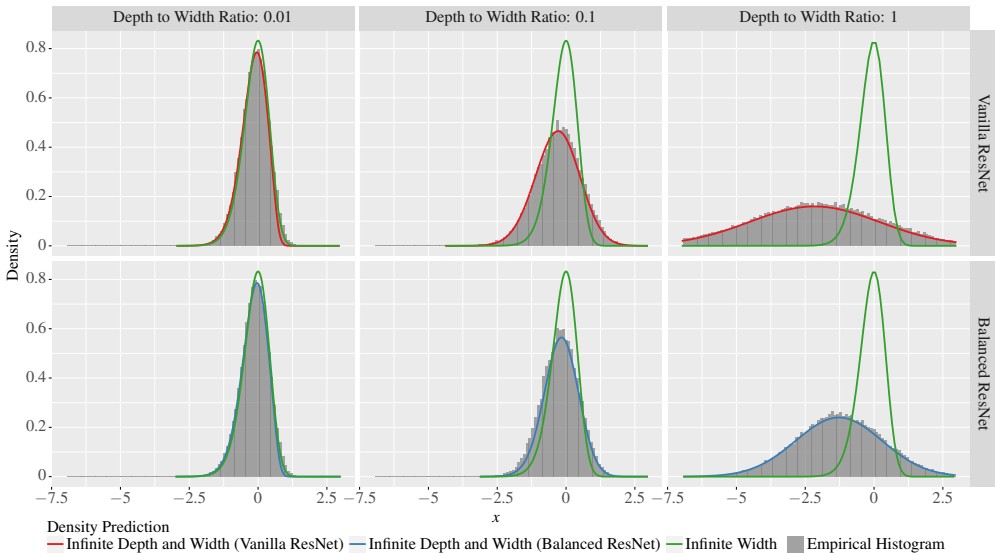

Figure 1: **Probability density function of** $\ln \|z^{\text{out}}\|^2$ for six network configurations on initialization. All networks have $n = 100, n_{\text{in}} = n_{\text{out}} = 10, \alpha = \lambda = 1/\sqrt{2}$. Depth $d$ varies by column. Top row: Vanilla ReLU ResNets. Bottom Row: Balanced ResNets (Section 2.1), which randomize the non-linearities used at each neuron thereby exponentially reducing the variance. The theoretical curves shown are the result of Theorem 1 and Theorem 4. These predictions converge to the infinite-width prediction when $d/n \to 0$. The details of the simulations and plot here is in Appendix C.

Despite the variety of these limits, one common feature is that the *depth* of the network (i.e., the number of its layers) is treated as a constant as the width of the network is allowed to grow. Indeed, for fixed width, $n$, the Gaussian approximations at initialization [18–23] worsen as the depth, $d$, increases. While real-world networks are fairly wide, their relative depth is not trivial. Hanin and Nica [24] were the first to compute an infinite-depth-and-width limit for fully connected networks. While the training dynamics of this limit are still not completely understood, we now know that, in the infinite-depth-and-width limit, the neural tangent kernel is random, and the derivative of the kernel is nonzero at initialization. This means training does not correspond to that of a linear model, like it does in the NTK limit [25, 26].

In addition to these theoretical corrections to the Gaussian limit, practitioners have begun to notice that even basic properties of standard neural networks do not line up with those predicted by the infinite-width limit. Perhaps the most basic is that the gradients of the network early in training are not Gaussian, but instead are approximately log-Gaussian [27]. In fact, one should see log-Gaussian behaviour agrees with the theoretical predictions of Hanin and Nica [24], although there has yet to be a careful empirical comparison made between the precise predictions coming from infinite-depth-and-width models and real world networks.

In practice, however, fully connected networks are not often used without architectural modifications. In particular, residual connections, ushered into widespread use after the description of the ResNet architecture [28], produced very deep architectures that were practically useful with optimization techniques available at the time. Initialization schemes for ResNets have been studied in the infinite-width limit [19, 29] or with modifications [30–32].

In this work, we consider the infinite-depth-and-width limit of fully connected architectures with residual connections ("Vanilla ResNets") and standard initializations. Here, the analysis is complicated by the effect of skip connections, which introduce interlayer correlations that have a non-negligible effect in this limit. Surprisingly, we observe a counter-intuitive but fundamental phenomenon, whereby these skip connection cause the network to be **hypoactivated**, meaning that less than half of the neurons are activated on initialization. This fact undermines key assumptions that underlie other infinite-depth-and-width limit studies of architectures without residual connections or with nonstandard modifications, such as post-activation residual connections.

Hypoactivation is a roadblock that all theoretical research into standard ResNet architectures must contend with—it is an unavoidable property of the standard architecture and the root of many technical difficulties. In order to sidestep this roadblock, we introduce a conjecture that bounds the size of hypoactivation and effect on interlayer correlation. The conjecture is inspired by empirical evidence from Monte Carlo simulations, as well as several simplified analyses that ignore certain technical difficulties. The conjecture introduces what we believe to be the minimal assumption necessary to allow rigorous theoretical work to proceed. It also defines an important open problem in the study of limits of residual architectures.

To demonstrate the utility of the conjecture, we show that it leads to precise predictions. In particular, we prove a limit theorem characterizing the exact marginal distribution of the output at initialization in the infinite-depth-and-width limit, up to order $O(dn^{-2})$. Our limit result shows that ResNets have *log-Gaussian behaviour* on initialization, and like fully connected networks [24], the behaviour is determined by the depth-to-width aspect ratio. This corroborates recent empirical observations about deep ResNets [27]. Since real world networks are finite, the question of how well this approximates finite behaviour is of paramount importance. Based on Monte Carlo simulations, we find excellent agreement between our predictions and finite networks (see Figure 1). Moreover, for very deep networks (e.g. $d/n = 1$) the infinite-depth-and-width prediction is extremely different than the infinite-width prediction. More surprisingly, however, is that even at comparatively small depth-to-width ratio (e.g. $d/n = 0.1$) the two limits are already significantly different. Furthermore, we also observe that the effects due to hypoactivation and interlayer correlation are non-negligible; these effects are precisely the difference between Vanilla ResNets and so-called "Balanced Resnets" in Figure 1. Perhaps most importantly, we observe that real network outputs exhibit exponentially larger variance than predicted by infinite-width limits. This type of variance at initialization is known to cause exploding and vanishing gradients and other types of training failures [33, 34]. Our result also implies that the output neurons of the network are not independent as predicted by the Gaussian infinite-width limit. See Figure 3.

In order to maintain the same skip connections from layer to layer, but render the activation patterns completely independent from layer to layer on initialization, we introduce the Balanced ResNet architecture, where the sign of each neuron's activation function is randomized. We demonstrate that this exponentially decreases the variance of the network output on initialization. Moreover, this independence between neurons also makes the model more amenable to theoretical analysis and opens the door to future understanding of network behaviour. Although it is beyond the scope of this paper, a small preliminary empirical study (Appendix C) suggests that standard training regimes are not negatively affected by replacing standard ResNet architectures with Balanced ones.

We summarize our main contributions as follows:

- We identify and characterize a fundamental property of ResNets that we call **hypoactivation**: less than half of the ReLU neurons are activated. Based on empirical evidence, we formulate a precise minimal conjecture bounding the effect of hypoactivation that permits us to make precise, rigorous estimates for other properties of ResNets.

- We prove a limit theorem which shows that the output of ResNets on initialization exhibits **log-Gaussian** behaviour with parameterization depending on the depth-to-width ratio $d/n$.

- We provide **empirical evidence** from Monte Carlo simulations showing our theory provides more accurate predictions for simple properties of finite networks compared to the predictions made by the Gaussian infinite-width limit.

- We introduce the ***Balanced ResNet* architecture**, which corrects the hypoactivation and variance due to layerwise correlation from Vanilla ResNets. We also prove that the output for this architecture is log-Gaussian on initialization with exponentially lower variance. This simple modification can be applied to *any* neural network that uses ReLU activations.

## 2   Main Results

In terms of the notation in Table 1, a **Vanilla ResNet** with fully connected first/last layers and $d$ hidden layers of width $n$ is defined by

$$z^0 := \frac{1}{\sqrt{n_{\text{in}}}} W^0 x, \quad z^\ell := \alpha z^{\ell-1} + \lambda \sqrt{\frac{2}{n}} W^\ell \varphi_+ \left( z^{\ell-1} \right) \text{ for } 1 \le \ell \le d, \quad z^{\text{out}} := \frac{1}{\sqrt{n}} W^{\text{out}} z^d. \tag{1}$$

| Notation | Description | Notation | Description | Table 1: Notation |
|---|---|---|---|---|
| $n_{\text{in}} \in \mathbb{N}$ | Input dimension | $n_{\text{out}} \in \mathbb{N}$ | Output dimension | |
| $n \in \mathbb{N}$ | Hidden layer width | $d \in \mathbb{N}$ | Number of hidden layers (depth) | |
| $\varphi_+(\cdot)$ | ReLU function $\varphi_+(x) = \max\{x, 0\}$ | $\varphi_-(\cdot)$ | "Domain Flipped" ReLU $\varphi_-(x) = \max\{-x, 0\}$ | |
| $\alpha \in \mathbb{R}$ | Skip connection coefficient | $\lambda \in \mathbb{R}^+$ | Feed-forward coefficient | |
| $x \in \mathbb{R}^{n_{\text{in}}}$ | Input | $W^0 \in \mathbb{R}^{n_{\text{in}} \times n}$ | Weight matrix at layer 0 | |
| $z^{\text{out}} \in \mathbb{R}^{n_{\text{out}}}$ | Network output | $W^{\text{out}} \in \mathbb{R}^{n \times n_{\text{out}}}$ | Weight matrix at final layer. | |
| $z^\ell \in \mathbb{R}^n$ | Neurons (pre-activation) for layer $1 \leq \ell \leq d$ | $W^\ell \in \mathbb{R}^{n \times n}$ | Weight matrix at layer $1 \leq \ell \leq d$ all weights initialized i.i.d. $\sim \mathcal{N}(0,1)$ | |

Note that factors of $\sqrt{2n^{-1}}$ in the hidden layer are equivalent to intializing according to the so-called He initialization [35]. Other intializations correspond to changing the coefficient $\lambda$. This setup is similar to that of "Stable ResNets" [29], where the infinite-width limit is studied.

In the infinite-depth-and-width limit, the intuition that half of the ReLU units are active (i.e. nonzero) because of symmetry is surprisingly not correct. We find that the following quantities play an important role. We define the **average hypoactivation (of layer $\ell$)** and the **total hypoactivation of the network** by

$$h_\ell := \mathbf{E}\left[\left\|\varphi_+\left(\hat{z}^\ell\right)\right\|^2\right] - \frac{1}{2}, \qquad h_{\text{total}} := \sum_{\ell=1}^d h_\ell, \tag{2}$$

where $\hat{z}^\ell := z^\ell / \left\|z^\ell\right\|$ and $\mathbf{E}$ means expectation over the choice of random network weights on initialization. The average hypoactivation is a measure of how many ReLU neurons are activated in layer $\ell$; $h_\ell = 0$ indicates roughly half of the neurons are active. Counter-intuitively, we observe that in a vanilla ResNet, $h_\ell$ is *negative* and $|h_\ell| = O(1/n)$, indicating slightly less than half the neurons are active.[2] After compounding over $d$ layers, the total hypoactivation is of order $h_{\text{total}} = O(d/n) = O(1)$ in the infinite-depth-and-width limit. As we will see, this effect has a non-trivial contribution.

At the same time, we also find the covariance between the activations of various layers does not vanish in the infinite-depth-and-width limit. This motivates the definition of the **total interlayer covariance correction**

$$I_{\text{total}} := \sum_{1 \leq \ell \neq \ell' \leq d} \mathbf{Cov}\left(2\left\|\varphi_+\left(\hat{z}^\ell\right)\right\|^2, 2\|\varphi_+(\hat{z}^{\ell'})\|^2\right). \tag{3}$$

As with the hypoactivation, skip connections cause this term to be non-trivial. We formulate Conjecture 5, which contains a precise encapsulation of the behaviour of $\hat{z}^\ell$ we observe.

**Conjecture 5 (Informal).** In expectation, the layers $\hat{z}^\ell$ can be approximated by uniform random variables from the sphere up to a relative error of $O(1/n)$.

The conjecture is well supported by Monte-Carlo simulations (see Figure 4). We provide a more detailed discussion and a precise statement of Conjecture 5 in Section 4. Assuming the conjecture holds, we prove a limit theorem about the distribution of $z^{\text{out}}$. Informally, this says that $z^{\text{out}}$ is **approximately a log-Gaussian scalar** times an independent Gaussian vector

$$z^{\text{out}} \approx \frac{\|x\|}{\sqrt{n_{\text{in}}}}\left(\alpha^2 + \lambda^2\right)^{\frac{d}{2}} \exp\left(\frac{1}{2}\mathcal{N}\left(-\frac{\beta}{2} + 2ch_{\text{total}}, \beta + c^2 I_{\text{total}}\right)\right)\vec{Z}, \tag{4}$$

where $\vec{Z}$ has iid $\mathcal{N}(0,1)$ entries, and $\beta$ and $c$ are defined by

$$\beta := \frac{2}{n} + \frac{d}{n} \cdot \frac{5\lambda^4 + 4\alpha^2\lambda^2}{(\alpha^2 + \lambda^2)^2}, \quad c := \frac{\lambda^2}{\alpha^2 + \lambda^2}. \tag{5}$$

The precise statement, including asymptotic error bounds, is as follows.

**Theorem 1.** *For any choice of hyperparameters $n_{in}, n_{out}, n, d, \alpha, \lambda$, and every input $x$, the output $z^{out}$ at initialization has a marginal distribution which can be written in the form*

$$z^{out} \overset{d}{=} \frac{\|x\|}{\sqrt{n_{in}}}\left(\alpha^2 + \lambda^2\right)^{\frac{d}{2}} \exp\left(\frac{1}{2}G\right)\vec{Z}, \tag{6}$$

---

[2]For a quantity $f = f(n,d)$ whose dependence on width and depth may be implicit, we use the notation $f = O(d^a/n^b)$ to mean that, for all choice of constants $\alpha, \lambda, r_-, r_+ > 0$, there exists a constant $C > 0$ such that $|f(n,d)| \leq Cd^a/n^b$ for all $d, n$ where $r_- < d/n < r_+$. This notation will allow us to state precise limit theorems when $d, n \to \infty$ with the ratio $d/n$ converging to a constant.

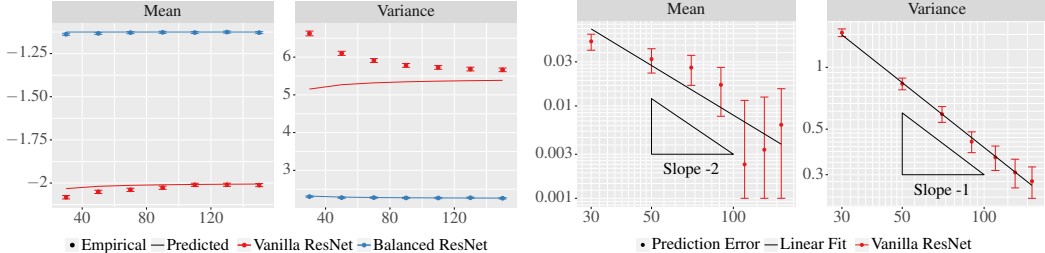

(a) **E** and **Var** of $G$ from (6) and prediction.  (b) Prediction error for **E** and **Var** of $G$ from (6).

Figure 2: Empirical mean and variance and infinite-depth-and-width prediction for the random variable $G(n, d, \alpha, \lambda)$ from (6) compared to the results of Theorems 1 & 4 . For these simulations, $\alpha = \lambda = 1/\sqrt{2}, d/n = 1$ are fixed and width $n$ is varied on the $x$-axis. The error bars indicate 95% confidence interval (CI) of the Monte-Carlo simulation, with truncation at $1e-3$ for plotting on log-scale. The balanced ResNet predictions fall within the Monte-Carlo CI and so are not plotted on log scale. Since $d/n$ is fixed, Figure 2b indicates the error in the asymptotic prediction of variance (7) is $O(n^{-1}) = O(dn^{-2})$ as claimed. On the other hand, the error in mean (7) is $O(n^{-2}) = O(dn^{-3})$; this is one order smaller than the statement proven in Theorem 1. A possible explanation is that the sub-leading error term in our approximation for $G$ is mean zero.

*where $\vec{Z} \in \mathbb{R}^{n_{out}}$ is a Gaussian random vector with iid $\mathcal{N}(0, 1)$ entries, $G = G(n, d, \alpha, \lambda)$ is a random variable which is independent of $\vec{Z}$ and whose distribution does not depend on $n_{in}, n_{out}$ or $x$.*

*Consider the limit where both the network depth $d \to \infty$ and hidden layer width $n \to \infty$ in such a way that the ratio $d/n$ converges to a non-zero constant. In this limit, assuming that Conjecture 5 holds, then the random variable $G \in \mathbb{R}$ has the following asymptotic behaviour:*

$$\mathbf{E}\left[G\right] = -\frac{\beta}{2} + 2ch_{total} + O\left(\frac{d}{n^2}\right), \quad \mathbf{Var}\left[G\right] = \beta + c^2 I_{total} + O\left(\frac{d}{n^2}\right), \tag{7}$$

*where $\beta$ and $c$ are as in (5), and moreover $G$ converges in distribution to a Gaussian random variable with mean and variance given by (7) in this limit.*

The main ideas of the proof of Theorem 1 is given in Section 5 and the detailed proof is given in Appendix B. We also provide more explicit formulas for $h_{\text{total}}, I_{\text{total}}$ below.

**Proposition 2.** *Assume Conjecture 5 is true. Then in the same infinite-depth-and-width limit as Theorem 1, the total hypoactivation $h_{total}$ and total interlayer covariance $I_{total}$ obey*

$$h_{total} = C_{\alpha,\lambda}\frac{d}{n} + O\left(\frac{d}{n^2}\right), \quad I_{total} = \sum_{1 \le \ell \neq \ell' \le d} \frac{\bar{J}_2(\theta_{|\ell'-\ell|}) - \bar{J}_2(\pi - \theta_{|\ell'-\ell|})}{n} + O\left(\frac{d}{n^2}\right). \tag{8}$$

*Here $C_{\alpha,\lambda}$ is a constant depending on $\alpha, \lambda$ and*

$$\bar{J}_2(\theta) \coloneqq J_2(\theta)/\pi = 3\sin(\theta)\cos(\theta)/\pi + (1 - \theta/\pi)\left(1 + 2\cos^2\theta\right),$$

*where $J_2(\theta)$ first appeared in [36], and $\theta_k$ is such that $\cos(\theta_k) = \alpha^k/(\alpha^2 + \lambda^2)^{k/2}$ .*

*Remark 3.* The Gaussian infinite-width limit predicts that the marginals of $z^{\text{out}}$ should have the form of (6) with $G$ being identically zero. As $z^{\text{out}}$ depends exponentially on $G$, the infinite-depth-and-width limit predicts the variance is exponentially larger than the infinite-width limit. See Section 3.1 for a detailed discussion and Figure 3 for verification against finite networks.

Using Monte Carlo simulations, we estimate the constant $C_{\alpha,\lambda}$. The result of Theorem 1 is then compared against finite networks in Figure 2. Full proof can be found Appendix B.

The idea of using a scaling parameter $\alpha = 1/\sqrt{2}$ in the skip connections has been noted in empirical papers [37] and also studied under simplified assumptions on the number of activation in each layer [38]. Our result shows that by choosing $\alpha^2 + \lambda^2 = 1$, the prefactor in (6) does not grow with depth thereby enabling deeper networks to be trained.

In the case that $\alpha = 0$, $\lambda = 1$, the architecture reduces to a fully connected network. In this case, log-Gaussian behaviour of the network was discovered in Hanin and Nica [24]. The fully connected case is simpler because the direction vectors $\hat{z}^\ell$ are always uniformly distributed on the unit sphere and are independent from layer to layer. This means $h_\ell = 0$ and $\theta_n = 0$ which greatly simplifies the result of Theorem 1.

Furthermore, the proof easily extends to the case where the coefficients $\alpha_\ell, \lambda_\ell$ vary from layer to layer; see Appendix A.1 for the general statement. Hayou et al. [29] and Hanin and Rolnick [33] have studied ResNets where $\alpha_\ell = 1$ in every layer, but $\lambda_\ell$ is allowed to vary. The prefactor of our result in this case becomes $\prod_{\ell=1}^{d} \left(1^2 + \lambda_\ell^2\right) \approx \exp\left(\sum_{\ell=1}^{d} \lambda_\ell^2\right)$, which is akin to the behaviour found in those papers. Additionally, our result precisely quantifies the log-Gaussian behaviour and its dependence on the sequence $\lambda_i$ through the parameters $\beta$ and $c$ in the infinite-depth-and-width limit.

## 2.1 Log-Gaussian Behaviour of Balanced ResNets

Given a collection of *iid uniformly random signs*, $s_i^\ell \in \{+, -\}, 1 \leq \ell \leq d, 1 \leq i \leq n$, a **balanced ResNet** is defined much like a Vanilla ResNet, except that a random sign is applied preactivation:

$$z^\ell := \alpha z^{\ell-1} + \lambda \sqrt{\frac{2}{n}} W^\ell \varphi_{s^\ell} \left(z^{\ell-1}\right) \text{ for } 1 \leq \ell \leq d,$$

where at each layer the vector function $\varphi_{s^\ell} : \mathbb{R}^n \to \mathbb{R}^n$ applies either $\varphi_+$ or $\varphi_-$ to the entries according to the random signs $s^\ell$. More precisely, the $i$-th component is

$$\varphi_{s^\ell}(z)_i := \begin{cases} \varphi_+(z_i) = \max(z_i, 0), & \text{if } s_i^\ell = +, \\ \varphi_-(z_i) = \max(-z_i, 0), & \text{if } s_i^\ell = -. \end{cases}$$

An equivalent definition is the entrywise multiplication $\varphi_{s^\ell}(z) = \varphi_+(s^\ell \odot z)$. Note that the random signs $s_i^\ell$ are *not* trainable parameters; they are frozen on initialization. This same symmetrization was first exploited by Allen-Zhu et al. [39] and Bai and Lee [40] to study a quadratic approximation of the network. We now present a corresponding limiting theorem for Balanced ResNets.

**Theorem 4.** *For a balanced ResNet, the same result as Theorem 1 given in* (6) *still holds, but with the mean and variance of $G$ given simply by*

$$\mathbf{E}[G] = -\frac{\beta}{2} + O\left(\frac{d}{n^2}\right), \quad \mathbf{Var}[G] = \beta + O\left(\frac{d}{n^2}\right). \tag{9}$$

Balanced ResNets are constructed so that the activation of each neuron is independent of all others due to the random signs $s^\ell$. This eliminates the hypoactivation and variance terms which complicated the analysis of the Vanilla ResNet and necessitated Conjecture 5. Instead, for Balanced ResNets it is straightforward to compute that for any fixed $z, w \in \mathbb{R}^n$ we have

$$\mathbf{E}\left[\|\varphi_{s^\ell}(z)\|^2\right] = \frac{\|z\|^2}{2}, \mathbf{Var}\left[\|\varphi_{s^\ell}(z)\|^2\right] = \sum_{i=1}^{n} \frac{z_i^4}{4}, \mathbf{Cov}\left[\|\varphi_{s^\ell}(z)\|^2, \|\varphi_{s^{\ell'}}(w)\|^2\right] = 0. \tag{10}$$

Even though the layers $\hat{z}^\ell, \hat{z}^{\ell'}$ are correlated, because the activation functions $\varphi_{s^\ell}$ and $\varphi_{s^{\ell'}}$ are set to be independent on initialization, the correlation between layers does not induce a correlation on which neurons are activated from layer to layer. This explains why there is no hypoactivation and interlayer correlation correction in Theorem 4 as there is in Theorem 1.

# 3 Consequences of Theorems 1 & 4 and Comparison to Infinite-Width Limit

## 3.1 Vanishing and Exploding Norms

By the basic fact $\mathbf{E}[\exp(\mathcal{N}(\mu, \sigma^2))] = \exp(\mu + \frac{1}{2}\sigma^2)$ it follows from Theorems 1 & 4 that, when the inputs $x$ has $\|x\| = \sqrt{n_{\text{in}}}$, the mean size scale of any neuron $z_i^{\text{out}}$ is approximately

$$\mathbf{E}\left[(z_i^{\text{out}})^2\right] \approx \begin{cases} (\alpha^2 + \lambda^2)^d \exp\left(2c h_{\text{total}} + \frac{1}{2} c^2 I_{\text{total}}\right), & \text{for Vanilla ResNets,} \\ (\alpha^2 + \lambda^2)^d, & \text{for Balanced ResNets.} \end{cases}$$

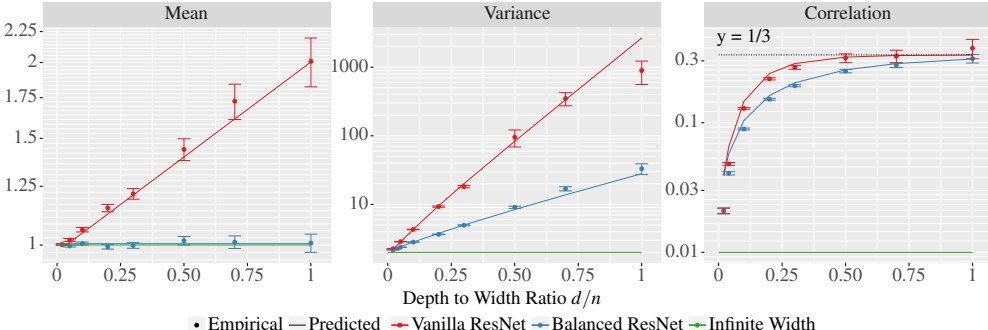

Figure 3: Behaviour of the mean and variance of the typical output $(z_i^{\text{out}})^2$ and the correlation between $(z_i^{\text{out}})^2, (z_j^{\text{out}})^2$ for two different output neurons. Here $n = 200$, $\alpha = \lambda = 1/\sqrt{2}$ and $d/n$ varies on the $x$-axis. The infinite width prediction for correlation is zero, but due to plotting on log-scale, we replace zero by $0.01$ for display.

(Note that the terms with $\beta$ cancel out!) When $\alpha^2 + \lambda^2 = 1$, this is constant for Balanced ResNets. In contrast, Vanilla ResNets have a complicated dependence on the network depth $d$ and width $n$ due to the *hypoactivation* and *correlations* terms. This means the behaviour is $\exp(Cd/n)$, which is somewhat surprising. A more serious issue is the variance which Theorems 1 & 4 predict to be

$$\mathbf{Var}\left[(z_i^{\text{out}})^2\right] \approx \begin{cases} (\alpha^2 + \lambda^2)^{2d}\left(3\exp\left(\beta + c^2 I_{\text{total}}\right) - 1\right)\exp\left(4ch_{\text{total}} + c^2 I_{\text{total}}\right), & \text{for Vanilla,} \\ (\alpha^2 + \lambda^2)^{2d}\left(3\exp\left(\beta\right) - 1\right), & \text{for Balanced.} \end{cases}$$

Since $\beta \approx Cd/n$, the term $\exp(\beta)$ represents exponentially larger variance for deep nets compared to shallow ones of the same width. In contrast, the variance predicted by the infinite width limit does not grow with depth for this model when $\alpha^2 + \lambda^2 = 1$. This effect means the relative sizes of different network outputs can be widely disparate. Unlike problems with the mean, this issue is harder to resolve. For example, normalization methods that divide all neurons by a constant does nothing to address the large relative disparity between two points. Techniques like batch normalization will be skewed by large outliers. This kind of variance is known to obstruct training [33]. The input-output derivative $\partial_{x_i} z^{\text{out}}$ has the same type of behaviour as $z^{\text{out}}$ itself; a simple proof is given in Appendix B. It is expected that the gradient with respect to the weights $\partial_{W_{ij}^\ell} z^{\text{out}}$ will also have the same qualitative behaviour [27] although more investigation is needed to understand this theoretically. Exponentially large variance for gradients is a manifestation of the vanishing-and-exploding gradient problem [34].

Balanced ResNets suffer from this variance problem less because the interlayer correlation term is zero. Since this variance reduction happens at the exponential scale in, the difference can be significant; for networks with $d/n = 1$, the contribution is a factor of $\approx e^{5.5} \approx 250$ for Vanilla ResNets vs. $\approx e^{2.5} \approx 10$ for Balanced ResNets. See Figure 3 for a comparison of these theoretically predicted properties vs experiments with finite networks.

### 3.2 Correlated Output Neurons

Since the same random variable $G$ multiplies the entire vector $z^{\text{out}}$, the individual neurons in the output layer are *not* independent. For example, Theorem 1 and 4 predict that the squared entries have strictly positive correlation given by $\mathbf{Corr}\left((z_i^{\text{out}})^2, (z_j^{\text{out}})^2\right) = (\exp(\sigma^2) - 1)/(3\exp(\sigma^2) - 1)$ for any two neurons $i \neq j$ where $\sigma^2 = \mathbf{Var}(G)$. This tends to $1/3$ as $d/n$ grows. The effect of correlated output neurons persists for Balanced ResNet but is reduced again due to the lower variance. This is very different from the infinite-width limit, which predicts that individual neurons should be independent Gaussians. This prediction of the theorem matches finite networks closely; see Figure 3.

## 4 Conjecture 5: Hypoactivation and Layerwise Correlations

Vanilla ResNets have a subtle asymmetry in the architecture due to the skip connections. Unlike fully connected networks, the distribution of $\hat{z}^\ell := z^\ell / \|z^\ell\|$ for $\ell \geq 1$ is *not* exactly uniformly distributed

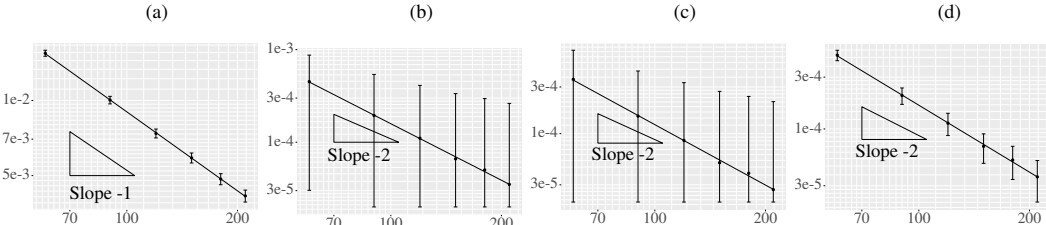

Figure 4: Monte Carlo evidence for Conjecture 5. $d = 200$, $\alpha = \lambda = 1/\sqrt{2}$, and $n$ varies on the $x$-axis. The plots show the quantities (a) $\left|\mathbb{E}\|\varphi_+(\hat{z}^\ell)\|^2 - \mathbb{E}\|\varphi_+(u)\|^2\right|$, (b) $\left|\mathbf{Var}\|\varphi_+(\hat{z}^\ell)\|^2 - \mathbf{Var}\|\varphi_+(u)\|^2\right|$, (c) $\left|\mathbf{Cov}\left(\|\varphi_+(\hat{z}^\ell)\|^2, \varphi_+(\hat{z}^{\ell-1})\|^2\right) - C(\theta_1)\right|$, (d) $\left|\mathbf{Cov}\left(\|\varphi_+(\hat{z}^\ell)\|^2, \varphi_+(\hat{z}^{\ell-2})\|^2\right) - C(\theta_2)\right|$. $C(\theta_k)$ is the theoretical covariance formula from the term where $\ell' - \ell = k$ in (8). Note also that the absolute error is expected to be $O(n^{-2})$ when the theoretical quantity is $O(n^{-1})$. Figures (a) and (b) verify the conjecture in (11) and (12), and Figures (c) and (d) verify (13) when $k = 1$ and $k = 2$ respectively. For display, we clip the bottom edge of the CI to 2e−5; otherwise the error bar would go down to $-\infty$ on the log scale.

on the unit sphere. Informally speaking, $\hat{z}^\ell$ can be thought of as a random walk whose variance at each step is proportional to $\left\|\varphi_+(\hat{z}^\ell)\right\|$. *More* randomness is injected when $\left\|\varphi_+(\hat{z}^\ell)\right\|$ is *large* and less when it is small. The net effect is that the walk moves slower when $\left\|\varphi_+(\hat{z}^\ell)\right\|$ is small, thereby spending more time in those locations. Consequently, $\|\varphi_+(\hat{z}^\ell)\|$ is biased toward smaller values.

The size of this effect is limited by entropy; most of the unit sphere $\mathbb{S}^{n-1}$ has $\|\varphi_+(u)\|^2 \approx 1/2$ in the sense that for any $\epsilon > 0$, the measure of the set $\left\{u : \left|\|\varphi_+(u)\|^2 - \frac{1}{2}\right| > 1/n^{\frac{1}{2}-\epsilon}\right\}$ vanishes as $n \to \infty$.

To quantify the effect of the bias, we can think of the evolution of $\hat{z}^\ell$, $\ell = 1, 2, \ldots$ as a random walk that takes different step sizes at a different points on the sphere. It is reasonable to expect that the behaviour of this processes will be similar to that of a time changed Brownian motion, which is slowed down at the points where $\hat{z}^\ell$ takes smaller steps. (Proving this comparison precisely is technically difficult since the parameter $n$ simultaneously plays both the role dimension and the step size of the walk.) Based on this heurstic comparison to time changed Brownian motion and on extensive Monte Carlo simulations we conjecture that the expected size of the hypoactivation effect is only $O(1/n)$ in expectation; Conjecture 5 contains a precise statement.

Even if each layer $\hat{z}^\ell$ is marginally close to the uniform distribution on the unit sphere, the directions $\hat{z}^\ell$ and $\hat{z}^{\ell+1}$ are not independent because of the skip connections in the network. As above, the exact behaviour is complicated due to fluctuations in the exact number of neurons which are activated in each layer. However, using the idea that $\left\|\varphi_+(\hat{z}^\ell)\right\|^2 = \frac{1}{2}(1 + o(1))$, we construct the following approximation. From (16), we have the approximation $z^{\ell+1}/\|z^\ell\| = \alpha\hat{z}^\ell + \lambda g^{\ell+1}/\sqrt{n}\,(1 + o(1))$ We observe that the norm of RHS is concentrated around $\sqrt{\alpha^2 + \lambda^2}$ as $n \to \infty$, so normalizing this to get $\hat{z}^{\ell+1}$ we have

$$\hat{z}^{\ell+1} = \left(\frac{\alpha}{\sqrt{\alpha^2 + \lambda^2}}\hat{z}^\ell + \frac{\lambda}{\sqrt{\alpha^2 + \lambda^2}}\frac{g^{\ell+1}}{\sqrt{n}}\right)(1 + o(1)).$$

Iterating this gives the same relationship for $\hat{z}^{\ell+k}$ where the first coefficient becomes $\alpha^k/\sqrt{\alpha^2 + \lambda^2}^k$. As before, based on Monte Carlo simulations, we conjecture that the size of the error is $O(1/n)$ in expectation. We formalize this as a precise statement in Conjecture 5 below.

**Conjecture 5.** *The distribution of the unit vector $\hat{z}^\ell = z^\ell / \|z^\ell\|$ is approximately uniformly distributed from the unit sphere $u \in \mathbb{S}^{n-1}$ in the precise sense that the following asymptotics hold*

$$\mathbf{E}\left[\left\|\varphi_+(\hat{z}^\ell)\right\|^2\right] = \mathbf{E}\left[\left\|\varphi_+(u)\right\|^2\right]\left(1 + O\left(\frac{1}{n}\right)\right), \tag{11}$$

$$\mathbf{Var}\left[\left\|\varphi_+(\hat{z}^\ell)\right\|^2\right] = \mathbf{Var}\left[\left\|\varphi_+(u)\right\|^2\right]\left(1 + O\left(\frac{1}{n}\right)\right), \tag{12}$$

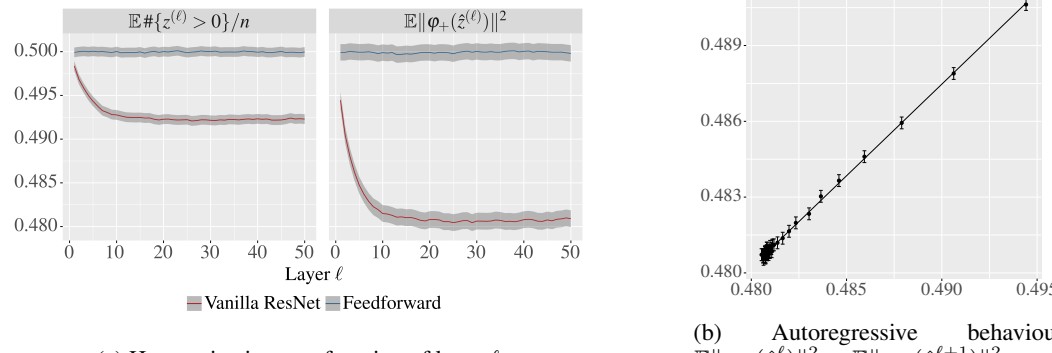

(a) Hypoactivation as a function of layer $\ell$

(b) Autoregressive behaviour $\mathbb{E}\|\varphi_+(\hat{z}^\ell)\|^2$ vs $\mathbb{E}\|\varphi_+(\hat{z}^{\ell+1})\|^2$

Figure 5: Monte-Carlo simulation for the behaviour of the unit vector $\hat{z}^\ell$ as a function of layer $0 \le \ell \le d$. Here $n = d = 50, \alpha = \lambda = \sqrt{2}^{-1}$. Figure 5a shows the mean fraction of neurons which are activated, $\mathbf{E}\left[\#\{i : z_i^\ell > 0\}\right]/n$ and the norm of the ReLU $\mathbf{E}\left[\|\varphi_+(\hat{z}^\ell)\|^2\right]$. The hypoactivation $h_\ell$ is how far this is from $\frac{1}{2}$. At layer $\ell = 0$, $\hat{z}^\ell$ is uniformly distributed from a unit sphere, but approaches a different steady-state as we go deeper into the network. Figure 5b shows evidence that the process $\mathbf{E}\left[\|\varphi_+(\hat{z}^\ell)\|^2\right]$ seems to be a linear function of the previous layer $\mathbf{E}\left[\|\varphi_+(\hat{z}^{\ell-1})\|^2\right]$. Figure 4 illustrates the dependence as $n$ varies.

*where the constants in the big $O(\cdot)$ notation are uniform in $\ell$. Moreover, for two layers $\ell, \ell'$, which are $k \ge 1$ layers apart $|\ell' - \ell| = k$, the joint distribution of $\hat{z}^\ell, \hat{z}^{\ell'}$ is approximately equal to the joint distribution of $u, \cos(\theta_k)u + \sin(\theta_k)g/\sqrt{n}$ where $g$ is a Gaussian vector with iid $\mathcal{N}(0,1)$ entries which is independent of $u$ and $\theta$ is such that $\cos(\theta_k) = \alpha^k/(\alpha^2 + \lambda^2)^{k/2}$ in the sense that the following asymptotics hold*

$$\mathbf{Cov}\left[\left\|\varphi_+(\hat{z}^\ell)\right\|^2, \left\|\varphi_+(\hat{z}^{\ell'})\right\|^2\right] = \mathbf{Cov}\left[\left\|\varphi_+(u)\right\|^2, \left\|\varphi_+\left(\cos(\theta_k)u + \frac{\sin(\theta_k)}{\sqrt{n}}g\right)\right\|^2\right]\left(1 + O\left(\frac{1}{n}\right)\right),$$
(13)

*where the constant in the big $O(\cdot)$ notation is uniform in $\ell, \ell'$.*

See Figure 4 for Monte Carlo simulations empirically verifying the conjecture for a fixed depth $d$, and see Figure 5 for verifying the uniformity in layers $\ell$. In particular, we observe that in Figure 5a, we can see the effect of hypoactivation converges rapidly to an equilibrium as the layer $\ell$ increases. In fact, we can further verify in Figure 5b that hypoactivation appears to be autoregressive, which implies the convergence is exponentially fast. This motivated the uniformity in layers in Conjecture 5.

## 5 Proof Ideas for Theorems 1 & 4

A key element of the proof is the following property of Gaussian random matrices. If $W$ which has iid $\mathcal{N}(0,1)$ entries, then for any vector $x$, we have

$$Wx \stackrel{d}{=} \|x\| g,$$
(14)

where $g$ is a vector whose entries are iid $\mathcal{N}(0,1)$ random variables. Because of the fully connected first and last layer of the network, (14) implies that

$$z^0 \stackrel{d}{=} \frac{\|x\|}{\sqrt{n_{\text{in}}}}g, \quad z^{\text{out}} \stackrel{d}{=} \frac{\|z^d\|}{\sqrt{n}}g'.$$
(15)

Hence $G := \ln\left((\|z^d\|^2/n)\cdot(\|x\|^2/n_{\text{in}})^{-1}\cdot(\alpha^2 + \lambda^2)^{-d}\right)$ only depends on $n, d, \alpha, \lambda$. (Equivalently, $G$ has the distribution of $\ln(\|z^d\|^2/n \cdot (\alpha^2 + \lambda^2)^{-d})$ when $z^0 = g$.) With this definition, (15) also shows $z^{\text{out}}$ is proportional to $\exp(G/2)$, establishing the first part of Theorem 1.

From this construction, the essence of the proof is to understand the distribution of $\left\|z^d\right\|$ when $z^0 = g$. To understand $\left\|z^d\right\|$, we look at the ratios $\left\|z^{\ell+1}\right\| / \left\|z^\ell\right\|$ layer by layer. By using the homogeneity property of ReLU $\varphi_+(|c|\,x) = |c|\,\varphi_+(x)$, we can divide $z^{\ell+1}$ from (1) by $\left\|z^\ell\right\|$ to obtain

$$\frac{z^{\ell+1}}{\|z^\ell\|} = \alpha\hat{z}^\ell + \lambda\sqrt{\frac{2}{n}}W^{\ell+1}\varphi_+(\hat{z}^\ell) \overset{d}{=} \alpha\hat{z}^\ell + \lambda\sqrt{\frac{2}{n}}\left\|\varphi_+(\hat{z}^\ell)\right\|g^{\ell+1}, \tag{16}$$

where $g^\ell$ are iid Gaussian vectors with iid $\mathcal{N}(0,1)$ entries by application of (14). Hence

$$\frac{\left\|z^{\ell+1}\right\|}{\|z^\ell\|} \overset{d}{=} \left\|\alpha\hat{z}^\ell + \lambda\sqrt{\frac{2}{n}}\left\|\varphi_+(\hat{z}^\ell)\right\|g^{\ell+1}\right\| \overset{d}{=} \left\|\alpha\vec{e}_1 + \lambda\sqrt{\frac{2}{n}}\left\|\varphi_+(\hat{z}^\ell)\right\|g^{\ell+1}\right\|.$$

The last equality follows by applying an orthogonal transformation $O$ such that $O\hat{z}^\ell = \vec{e}_1 = (1, 0, 0 \ldots, 0)^T$ inside the norm, and observing that Gaussian random vectors are invariant under orthogonal transformations $Og^\ell \overset{d}{=} g^\ell$. Hence we have the telescoping product for $\left\|z^d\right\|$:

$$\left\|z^d\right\| = \left\|z^0\right\|\prod_{\ell=0}^{d-1}\frac{\left\|z^{\ell+1}\right\|^2}{\|z^\ell\|} \overset{d}{=} \left\|z^0\right\|\prod_{\ell=0}^{d-1}\left\|\alpha\vec{e}_1 + \lambda\sqrt{\frac{2}{n}}\left\|\varphi_+(\hat{z}^\ell)\right\|g^{\ell+1}\right\|. \tag{17}$$

This shows that $\left\|z^d\right\|$ is a product of $d$ random variables which are dependent on each other only through the terms $\left\|\varphi_+(\hat{z}^\ell)\right\|$. (Note that $\left\|z^0\right\|$ is independent of $\hat{z}^0$ since $z^0$ is Gaussian.) Since $\left\|\varphi_+(\hat{z}^\ell)\right\|^2 \approx 1/2$ with typical fluctuations on the scale $1/\sqrt{n}$, therefore the dependence between terms of (17) is small.

Taking the $\ln$ of (17) exhibits $\ln(\left\|z^d\right\|^2/n)$ as a sum of these weakly correlated random variables. Here we note that various tail estimates for the same or related quantities have been developed [3, 41], however these estimates are not precise enough to pinpoint the exact limiting distribution. In contrast, we are able to derive the exact limiting distribution via a Central Limit Theorem (CLT) for weakly correlated sums [42]. The proof of Theorem 1 is completed by computing the mean, variance and covariance of terms using Conjecture 5. For Theorem 4, the final calculation is simplified by (10) which shows the terms are uncorrelated. The detailed proof is given in Appendix B.

## Acknowledgement

We would like to thank Blair Bilodeau, Gintare Karolina Dziugaite, Mahdi Haghifam, Yani A. Ioannou, James Lucas, Jeffrey Negrea, Mengye Ren, and Ekansh Sharma for helpful discussions and draft feedback. We would also like to thank the anonymous NeurIPS reviewers for insightful feedback. In particular, one identified numerous relations to existing work, and another helped us identify the uniformity requirement in Conjecture 5. ML is supported by Ontario Graduate Scholarship and the Vector Institute. MN is supported by an NSERC Discovery Grant. DMR is supported in part by an NSERC Discovery Grant, Ontario Early Researcher Award, and a stipend provided by the Charles Simonyi Endowment.

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
