# OpenReview forum: "The future is log-Gaussian: ResNets and their infinite-depth-and-width limit at initialization"
_NeurIPS.cc/2021/Conference — NeurIPS 2021 Poster_

### Official Review · Reviewer_Kh9k · 2021-07-08

**Rating:** 5
**Confidence:** 5

**Summary:**

This paper aims to deal with the problem that the gaussian process approximation of Neural network at initialization will become worse while the network become deeper due to the error accumulation. This paper proposed a central limit theorem in depth to have a log-gaussian limit of the infinite width and depth limit.

The subject of paper is pretty cool, but the assumption in the paper and conclusion doesn't convinced me the new characterization is right and essential.

**Limitations And Societal Impact:**

See main review


Additional point：
Can authors explain the relationship between your paper and [1]
I understand [1] is an ICML2021 paper and appears after Neurips deadline thus I'll not take the answer of this question into the decision

[1] On the Random Conjugate Kernel and Neural Tangent Kernel http://proceedings.mlr.press/v139/hu21b/hu21b.pdf


**Main Review:**

This paper is actually interesting although I provide a weak rejection decision, this paper

+ Provide a new limit of deep resnet and verified via experiment


Major points:

1. the (\sqrt{lambda^2+sigma^2}^d) in the limit will goes to the infinity which is not a good way to claim the convergence, you'de better to normalize it at left hand side, i.e. Z^{out}/\sqrt{lambda^2+sigma^2}^d convergence to ....

In the realistic resnet, we never use the lambda their, i.e. the (\sqrt{lambda^2+sigma^2}^d) will goes to infinity. If we consider a Neural ODE limit the sigma will also depends on the depth, as an example \sqrt{lambda^2+1/3}^d --> constant sounds a better limit.

2.Why the log-gaussian process matters for understanding ResNet? I can't find evidence in the paper.

For the empirical paper you cited they said the gradient is log-gaussian but not the features, here is still a gap for the backward process is strongly correlated with the forward one, how the CLT can be used is not obvious to me.


3. I'm not satisfied with the assumption, the O(1/n) error in assumption is uniformly on depth in my mind? I would like the author to clarify this and provide intuition why this can hold?


4. Can the assumption be proved via induction like Greg Yang's tensor program papers?

5.We know that infinitely wide NN is a gaussian **process**, this paper only tells us what's the feature distribution of a single data point but failed to answer what's the process. Can the proof generalize to the off-diag term in the correlation matrix of the process?
I think it relates to quesiton2. If you add a layer norm before the last layer, bot log-gaussian and gaussian will go back to a uniform distribution on the sphere. I wonder whether this paper is playing on the scaling. I think investigating off-diag term can be much helpful



Minor points:
1.I can't see the proposed benefit of the balanced ResNet in practice, if this part is deleted, I think no information will lost in my mind.

2.For the experiment, if you can show keep n/d as a constant and the limiting distribution is not changing can be interesting.


3.although the proof is not hard, it takes some time to follow it. I hope the author can take some time to organize the prove. [the logic of the proof and lemmas can be put in a section and the detailed calculation of variance can be put later

Other question doesn't relate to the decision
To deal with deeper network [1] and their recent book provide a finite depth correction to do this, can their approaches gives a log-gaussian?


[1] B. Hanin and M. Nica. “Finite Depth and Width Corrections to the Neural Tangent Kernel.”
Int. Conf. Learning Representations (ICLR). 2019.



I believe this paper leads to an interesting direction, but at this point I still find out the paper can't convince me of the conclusion given.

=======================================

After rebuttal the reviewer still doesn't convinced me the reason why the assumption is not strong, thus I changed my score to 4.

I think the author need time to
- distinguish the difference and relationship with the previous techniques
- clarify why I can trust the assumption is uniform in d

**Time Spent Reviewing:**

1.5

---

> ### Author Response · Authors · 2021-08-10
> **Response to Reviewer Kh9k**
>
> # Summary
>
> We thank the reviewer for the comments and interest in our work. The review begins with “this paper is actually interesting,” but raises five “major” points. One of these points is actually asking about a potential proof technique to resolve the conjecture (unfortunately, we don't think this works); two points are raising technical concerns (both are, luckily, misconceptions that we resolve below, and that will help improve the paper because they are likely to be common ones); and two are questions about significance, which we address below, by arguing that the infinite-depth-and-width setting is **much** more complex than the infinite-width setting, that one cannot expect a simple Gaussian limit that fully characterizes ResNets at initialization. Furthermore, our results are the first step towards a tractable theory for finite size ResNets (Figure 1 and 3 already showing promise), but that the task is going to be one solved by a community of scientists working together, not one paper.
>
> We hope our response will satisfy the reviewer and contribute towards an improved score for our submission.
>
> # On the Potential Proof of Conjecture
>
> The reviewer asked if Conjecture 5 can be proved via Greg Yang's tensor programs approach. Unfortunately, tensor programs can only handle the infinite-width regime, which cannot analyze the accumulation of $O(1/n)$ size correction terms over a large number of layers, and the conjecture is regarding this type of error accumulation. To our best knowledge, infinite-width analyses typically fall prey to this issue. This is one of the main reasons we consider our contribution significant and novel.
>
> # On the Normalization Constant
>
> Here we would like to respond to the first major point on the normalizing constant. Here the reviewer is confusing the statement of Theorem 1, where eqn. (6) is literally just rewriting the network output formula, where $G$ is the “normalized” random variable of interest that we provide a convergence result on. We wrote the dependence explicitly to point out that by not including a factor of $\alpha$, the term $(\alpha^2+\lambda^2)^d$ will explode if left unchecked, which is often neglected in existing work. We provide a detailed discussion after Remark 3.
>
> # On Conjecture 5
>
> The reviewer raised a question regarding how the conjecture can hold uniformly in depth $d$. The easiest way to see this is to observe Figure 9, where we plot the hypoactivation effect as a function of depth. By viewing depth as a time variable, we can see that hypoactivation converges exponentially fast to an equilibrium, where the dynamics is very close to autoregressive processes. Therefore, the depth $d$ will have minimal effect on hypoactivation once it is larger than ~20. Furthermore, as we increase the depth-to-width ratio in Figure 1, the prediction remains an indistinguishably close match to finite networks, which gives further credence to the uniformity in depth.
>
> # On the Significance of Our Results
>
> Here we will first address the second point on the significance of the log-Gaussian distribution for studying ResNets at initialization. By observing Figures 1 and 3, we can see that our log-Gaussian results provide a significant improvement in predicting the output distribution over infinite-width limits. This shows the promise of infinite-depth-and-width limits for modeling finite networks, and provides an important starting point to studying the complete problem.
>
> The reviewer also suggested that we study the “off-diagonal term”, i.e. joint distributions of two inputs. Unfortunately, due to the inherent complexity of the infinite-depth-and-width limit, we do not have a nice limit as opposed to the infinite-width case. In fact, our on-going work suggests that the joint distribution over $k$-inputs cannot be characterized by pairwise dependence structure alone; in other words, the joint distribution won’t look like a Gaussian process, but rather a more general nonparametric distribution. Therefore, resolving the joint distribution problem will require a significant improvement in technical tools that are currently unavailable.
>
> Ideally we would like to solve the joint distribution before submitting, but as the reviewer mentioned that since the NeurIPS deadline, multiple related work [1,2] have surfaced on the same topic. As this is a fast moving field, we want to report frequently on our progress so we don’t miss opportunities to contribute to the community. Given that we are already far exceeding the page limit of NeurIPS and introduced significant technical progress compared to existing work, we felt the current scope of the paper is appropriate for submission.
>
> # Other Comments
>
> On proof organization, the reviewer raised a concern regarding the order of the logic and calculations. We would be happy to accept suggestions on improving the presentation of the proof, however we already have the logic contained in a proof sketch section, and the variance calculation is contained only in Section B.2. Is the reviewer suggesting we move this section later?
>
> The reviewer also commented that the conclusion is unconvincing. Could the reviewer elaborate on what specifically is not convincing? We would be happy to clarify and add to any existing theoretical or empirical content in the paper.
>
> # On Related Works
>
> The reviewer also asked about the connections to [3] and presumably the recent book refers to [1]. Indeed the path counting techniques of [3] are based on [4], which proved a similar log-Gaussian result for feedforward networks. However, we would like to point out that this path counting technique cannot handle hypoactivation and interlayer correlation effects, whereas our approach drastically simplifies the calculations required. On the other hand, [1] studies a finite depth and width correction based on a Taylor expansion approach, which allows them to compute in closed form first order corrections to training dynamics and generalization properties to the infinite-width regime. However, as this approach only studies a truncation of the Taylor series in terms of the depth-to-width ratio, this will not lead to a log-Gaussian distribution. While our approach leads to a clean proof of the log-Gaussian limit, each of these techniques have their respective advantages and disadvantages, and should be considered for corresponding applications.
>
> The reviewer also asked us to comment on [2], which was available after the NeurIPS submission deadline. Indeed [2] proved a similar log-Gaussian result to ours, however, we would like to point out that the authors decided to study a ResNet architecture that added skip connections after ReLU activations, which is known to perform worse in practice [5]. This subtle change is the root cause of hypoactivation and interlayer correlations, which [2] did not have to handle. That being said, our balanced ResNet theorems can be immediately tweaked to handle this case.
>
> [1] https://arxiv.org/abs/2106.10165
>
> [2] http://proceedings.mlr.press/v139/hu21b/hu21b.pdf
>
> [3] https://arxiv.org/abs/1909.05989
>
> [4] https://arxiv.org/abs/1812.05994
>
> [5] https://arxiv.org/abs/1603.05027

---

> > ### Comment · Reviewer_Kh9k · 2021-08-10
> > **Still not justify my concerns**
> >
> > I read the rebuttal carefully but rebuttal still doesn't justify my concerns
> > ## On Conjecture 5
> > I understand the accumulation of  O(1/n) size correction terms over a large number of layers [d \proto n times] will become problematic when using the Tensor program. But as mentioned in your rebuttal, "By viewing depth as a time variable, we can see that hypoactivation converges exponentially fast to an equilibrium", the exponential convergence of the iterative process may lead the term to be controllable. I think the author still can't convince me of this term.
> > The proof should look like
> > - the iterative process of the dynamic is stable --> exp convergence as an example
> > - plugin tensor program
> > - using a gronwall like inequality to bound the correction terms to be O(1/n)
> >
> > At the same time, the uniformness of O(1/sqrt n) in depth is not mentioned in the main text (correct me if I'm wrong) and the reason why the uniform of O(1/sqrt n) in depth is reasonable is also missing in the current draft. I think it's essential for the reader to understand the hardness of the conjecture and what it's meaning exactly.
> >
> >
> > ## On the Significance of the Results
> > I'm not sure the author consider which of the following part as the significance of their results
> > - propose the log-gaussian limit
> > - the proof the log-gaussian limit
> >
> > If consider the first one as the main contribution, the relationship with related work and the significance of the log-gaussian approximation will become crucial. (why I need log-gaussian, will it affect my prediction result (I mean the classification result, but not the feature, the un-learned feature is boring) or it's just playing on the scaling/magnitude of mean and variance.)
> > As an example, [joint distribution over -inputs cannot be characterized by pairwise dependence structure alone] is interesting, but it's far beyond this paper's results.
> >
> >
> >
> > If consider the second one as the main contribution, conjecture 5 is problematic. Conjecture 5 is the hardest part in my mind, the other part is much easier.
> >
> > ## On Related works
> >
> > In my mind, the path counting technique can handle hypoactivation and interlayer correlation effects, it's just changing an objective to calculate. ResNet are not products of independent random matrices, they are products of (I+random matrix) and still can decompose to different paths. I think they did this in the recent book(arXiv:2106.10165).
> > I'm happy to hear about specific reasons why the path counting technique can't handle this objective.
> >
> > I agree your approach drastically simplifies the calculations required, but they don't need Conjecture 5 in my mind. The comparison is unfair at this point to me.
> >
> > Besides, (arXiv:1603.05027) is a after training result, the paper deals with the initalization. If you are saying [2]'s network is not good at initialization, I think you need experiment to support it.
> >
> > ## On the normalizing factor
> >
> > What I'm saying is if you write it down in the way "Z^{out}/\sqrt{lambda^2+sigma^2}^d convergence to ....", it's more precisely and rigorously in mathematics. In mathematics, the theorem you have now they are not converging in the epsilon-delta definition...
> >
> > I know the scaling is like this and maybe other papers write it down like this, but I think as a rigorous mathematical theorem, you should write it down properly.
> >
> > If changes [2] to your network, I still believe their proof can gives the asymptotic distributions.
> >
> > ## Other Points
> > I also have a comment in the original review:
> > For the empirical paper you cited they said the gradient is log-gaussian but not the features, here is still a gap for the backward process is strongly correlated with the forward one, how the CLT can be used is not obvious to me.
> > The correlation between z^out and the weight is a hard thing in my mind, however in tensor program paper, they claimed they are indepenedent in the infinite width limit.
> >
> > It seems not answered
> > B. Chmiel, L. Ben-Uri, M. Shkolnik, E. Hoffer, R. Banner, and D. Soudry. “Neural gradients are near-lognormal: improved quantized and sparse training.” Int. Conf. Learning Representations (ICLR). arXiv: 2006.08173. URL: https://openreview.net/forum?id=EoFNy62JGd.
> >
> >
> >
> > ## Overall
> > I think the general idea of the paper is interesting, but I think the author still need modification to the draft. Everyone knows arguing that the infinite-depth-and-width setting is much more complex than the infinite-width setting and this paper aims to answer the question. But as a theory paper, the paper has too many unrigorous claims that should be improved.
> >
> >
> > The rebuttal is too defensive in my mind, the reviewer is here to help for a better version but not the enemy for rejecting/accepting.

---

> > > ### Author Response · Authors · 2021-08-13
> > > **Response to Reviewer Kh9k**
> > >
> > > # Summary
> > >
> > > We thank the reviewer for the additional response. The reviewer clarified on the previous concerns raised, and summarized the main issue as “the paper has too many unrigorous claims.” Similar to reviewer SxtD, we believe at the core of the reviewer’s concerns is the technical contributions in the proof of the balanced ResNet result (Thm. 4), which is unconditional on any assumptions. Therefore we will begin by clarifying our technical contributions in comparison to existing feedward results [1]. Afterwards, it will be clear that Conjecture 5, which is the **only** unproven (but empirically verified claim), is identifying the final technical cornerstone required to resolve the full problem. In fact, we believe that formulating the precise conjecture is itself a contribution, as the hypoactivation effect was unknown prior to our paper. Furthermore, we emphasize that our results are already sufficient to make an accurate prediction for the output density, as illustrated by Figure 1.
> > >
> > > Again, we reiterate our desire to resolve the full problem before submitting. However, as the reviewer mentioned, since the NeurIPS deadline, multiple works [2,3] have surfaced on the same topic. Therefore to avoid missing out on contributing to the community, we would like to report frequently on the technical progress we have made and precise formulations of interesting new problems (Conjecture 5), so as to help the entire field move forward together.
> > >
> > > We hope our further response will be more convincing to the reviewer, and help contribute towards an improved score.
> > >
> > > # On Technical Novelty Beyond Feedforward Networks
> > >
> > > The reviewer raised a question regarding what we considered contributions of the paper.
> > >
> > > To start, we introduced a new approach by analyzing layer norms and its variances for the Balanced ResNet (Thm. 4), which is already a non-trivial calculation as also suggested by reviewer xrqc. As we mentioned before, this approach also allows us to compute the effects of hypoactivation and interlayer correlation, whereas the path counting arguments of [1] cannot handle this complication (to be discussed further).
> > >
> > > Once we recognize the technical novelty up to this point, we will see that hypoactivation is the only remaining problem. Instead of writing it off as a technical assumption to avoid the difficulty, we identified and narrowed down the new phenomenon, provided a precise conjecture, and verified it via extensive Monte Carlo simulations. Furthermore, if we observe the statement of Thm. 1, the only effect hypoactivation has is shifting the mean of $G$, whereas the interlayer correlation effects are already characterized by Prop. 2. This shift in the mean can then be precisely estimated via further Monte Carlo simulations (Figure 7).
> > >
> > > Therefore, Conjecture 5 is not the main technical difficulty, but rather filling in the role of identifying the final problem, to which we have an accurate Monte Carlo verification of. We again emphasize that the conditional results of Thm. 1 and Prop. 2 are already sufficient to predict the output density, which is then plotted in Figure 1, showing a very accurate match to finite size ResNets.
> > >
> > > To our best knowledge, this is the first work to provide an accurate density prediction for ResNets at initialization, which provides a foundation to which we can then study further interesting properties, such as the joint distribution of output (and eventually NTK) over multiple inputs at initialization.
> > >
> > > # On Path Counting Arguments under Hypoactivation
> > >
> > > The reviewer asked for a clarification on why the path counting arguments of [1] cannot handle ResNets. The key simplification [1] relies heavily on is replacing each layer’s ReLU activations with an independent diagonal random matrix, where the entries are Bernoulli random variables. However, in the ResNet setting, these diagonal matrices will no longer be independent from each other. If we continue to study the path decomposition using the same approach, we will find that even computing the moments of a single path is non-trivial due to the dependence structure in the Bernoulli random variables.
> > >
> > > For additional intuition, we can consider the distribution of the Bernoulli random variable $\xi := 1_{ z^{\ell+1}_i > 0 }$ conditioned on the previous layer $z^\ell$. Observe that whenever $z^\ell_i \neq 0$, we have the Bernoulli parameter $p := \mathbb{P}(\xi = 1)$ will not be $1/2$. Furthermore, $p$ will depend on the value of the entire vector $z^\ell$. This implies that to compute the moments of a single path, one would need to analyze the dependence structure on neurons **outside of the path**. Consequently, even the most basic component of a path counting argument would fail here.
> > >
> > > We would also like to emphasize that path counting does not work around Conjecture 5. To continue under this approach, one would need to resolve an equivalent technical roadblock caused by hypoactivation.
> > >
> > > # On the Normalizing Factor
> > >
> > > The reviewer reiterated the concern regarding the presentation of Thm. 1 as unrigorous. We believe this comment is based on a misunderstanding of the statement in Thm 1: specifically confusing $Z^{\text{out}}$ with $G$. We are **not** claiming the convergence of $Z^{\text{out}}$ directly in Thm 1. but rather we prove a rigorous convergence result for $G$. And indeed, as the reviewer points out, $Z^{\text{out}}$ will diverge when $\alpha^2+\lambda^2 > 1$, which is often neglected in existing work.
> > >
> > > The reviewer suggested we normalize $Z^{\text{out}}$ by $(\alpha^2 + \lambda^2)^d$,and indeed this factor appears in the definition of the random variable $G$. For convenience, we will copy the formula for $G$ from line 244 below:
> > >
> > > $$ G := \ln \left( \frac{ \|z^d\|^2 }{ n } \frac{ n_{in} }{ \|x\|^2 } \frac{1}{ (\alpha^2 + \lambda^2) } \right) ,
> > > $$
> > >
> > > which is a (random) function of $n, d, \alpha, \lambda$, and Thm. 1 provides a rigorous convergence result for this variable as $d,n \to \infty$.
> > >
> > > (Note: We assume the reviewer means $\alpha^2+\lambda^2$ and not $\lambda^2+\sigma^2$ as written in this comment, as there is no $\sigma$ in the paper. )
> > >
> > > # On The Tensor Programs Proof of Conjecture 5
> > >
> > > We would appreciate a more detailed explanation of how to plugin the tensor program to prove Conjecture 5. While we are not experts on the topic, we understand tensor programs as an asymptotic theory, which does not characterize $O(1/n)$ error terms explicitly.
> > >
> > > At the same time, we do not understand how we can use a Gronwall-like inequality to control the cumulative error. To achieve a vanishing error term via Gronwall, you will need each layer to contribute at most $O(1/n^{1+\epsilon})$ error from the limit. In our case, while the hypoactivation process converges to an equilibrium, this equilibrium itself contains an $O(1/n)$ error from the uniform distribution on $\mathbb{S}^{n-1}$. Therefore, using a Gronwall-like inequality will introduce an error term of size $O(d/n)$ over $d$ layers, hence not vanishing in the infinite-depth-and-width limit.
> > >
> > > Furthermore, Figure 1 provides empirical evidence that the cumulative error due to hypoactivation does not vanish. In particular, the predicted density (in blue) for the balanced ResNet is exactly the density without correcting the mean for hypoactivation (and variance for interlayer correlation). The prediction would be significantly off from the empirical distribution, i.e. if we plot the blue curves in the top panels, it would be far from the grey histograms, even in mean alone.
> > >
> > > # On Related Work
> > >
> > > The reviewer questioned how we can use the CLT for computing the distribution of the gradient, as [4] showed empirically that gradient magnitudes appear to be log-normal distributed. We are unsure which part of our work the reviewer is referring to, as we did not claim any technical results regarding gradients, but we merely cited the empirical work of [4] to discuss how our work fits into the existing literature.
> > >
> > > The reviewer also raised several further questions regarding [2,3], both of which the reviewer claimed will not factor into the decision, as these only surfaced after the NeurIPS deadline. We will however address the questions below.
> > >
> > > Firstly, the reviewer commented that “I think they did this in the recent book” referring to a path counting argument for ResNets in [3]. We do not believe the random matrix theory kind of path counting method from [1] appears in this book. A different kind of counting using Wick’s theorem for moments is used in this book.
> > >
> > > Next the reviewer asked if we are claiming “[2]'s network is not good at initialization,” as [5] compares the performance of pre/post-activation skip architectures after training. We think the reviewer misunderstood our point, as we simply wanted to say there is additional motivation to studying the pre-activation skip architecture shown by [5].
> > >
> > > Finally, the reviewer claimed “If changes [2] to your network, I still believe their proof can gives the asymptotic distributions.” While we are not experts in the techniques used by [2], their architecture will not observe hypoactivation effects, as all inputs to ReLU will be symmetric, i.e. their directions are uniformly distributed on the sphere. Therefore, even if one were to use the approach of [2], Conjecture 5 will still need to be resolved, and hence unable to recover the output distribution without additional technical novelty.
> > >
> > > # References
> > >
> > > [1] https://arxiv.org/abs/2106.10165
> > >
> > > [2] http://proceedings.mlr.press/v139/hu21b/hu21b.pdf
> > >
> > > [3] https://arxiv.org/abs/1909.05989
> > >
> > > [4] https://openreview.net/forum?id=EoFNy62JGd
> > >
> > > [5] https://arxiv.org/abs/1603.05027

---

> > > > ### Comment · Reviewer_Kh9k · 2021-08-13
> > > > **Quick Response**
> > > >
> > > > ## On The Tensor Programs Proof of Conjecture 5
> > > > Yes, If you simply accumulate the error,it'll become $O(d/n)$
> > > >
> > > > But the procedure I propose will like this, every layer will introduce a new $O(1/n)$ error. If we denote the error for the feature distribution to the uniform distribution is $\rho_L$, can we have $\rho_L< \alpha \rho_{L-1}+O(1/n)$, then I can do the grownwall.
> > > >
> > > > This process is a way to make what you said in the rebuttal rigorously. You claimed that the reason you believe conjecture 5 can hold uniformly in depth is that "hypoactivation converges exponentially fast to an equilibrium empirically". If the process can't hold, I can't see the reason why "hypoactivation converges exponentially fast to an equilibrium empirically" can lead to uniformly hold in depth.
> > > >
> > > > I understand in figure 1 the Gaussian process approximation is bad. I think I need to correct my words. What I meant by the tensor program is not the propagation of the mean and covariance of the Gaussian process. What I meant is to calculate the distribution of the feature, you can use another series of Gaussian random variables to approximate in terms of the infinite width limit. In this case, you can have the same central limit theorem in depth as your prove had.
> > > >
> > > > ## On Normalizing Factor
> > > > The G definition appears in line 244, but the main theorem is claimed in line 128. I don't think theorem 1 is claimed as you claimed in the rebuttal. My request is only to change the way to describe Theorem 1.
> > > >
> > > > You said your theorem is formulated in the G defined in line 244, where is it? It's not theorem 1 right?  The theorem 1 is Z^out converge to another independent distribution right?
> > > >
> > > > I can't understand the author is so defensive on this point. It's an easy change.
> > > >
> > > >
> > > >
> > > > ## On Path Counting Arguments
> > > > I'll read the Path Counting Arguments papers carefully the next day to see whether they can be used here. At this time, the rebuttal doesn't include any specific reason why it can't be used. only marked a "(to be discussed further)."
> > > >
> > > > If the author provides a specific reason, it could help.

---

> > > > > ### Comment · Reviewer_Kh9k · 2021-08-22
> > > > > **Unfair calim about the path counting arguments**
> > > > >
> > > > > After reading [1] these days, I strongly disagree the author's claim "path counting arguments can't work for resnet". [1] showed us how it can be adapted for ResNets. Yes [1] isn't the reason I don't come to accept the paper, but caliming "path counting intractable" in rebuttal will not help to increase the score. I hope the author should have a fair discussion about the "path counting argument" in the next version.
> > > > >
> > > > > I hope the author provides me evidence about the reason why I can trust the conjecture can hold **uniformly in d** as I said in the previous comment and response about the normalizing factor at the same time.
> > > > >
> > > > > At this point I temporally  changed my score to 4 until the author response the questions.
> > > > >
> > > > >
> > > > > [1] On the Random Conjugate Kernel and Neural Tangent Kernel http://proceedings.mlr.press/v139/hu21b/hu21b.pdf

---

> > > > > > ### Author Response · Authors · 2021-08-23
> > > > > > **Response to Reviewer Kh9k**
> > > > > >
> > > > > > # On Path Counting and Comparison with Hu and Huang [1]
> > > > > >
> > > > > > We think we might be misunderstanding you. Would it be possible to get clarification on a few points around path-counting?
> > > > > >
> > > > > > If we understand, you are asking us to provide more justification for our claim that "there are challenges applying path counting arguments to residual networks". To begin, we think it is important to clarify that, when we say "path counting", we are referring to the techniques used by Hanin and Nica [2,3].
> > > > > >
> > > > > > ## Hu and Huang [1] don't use path-counting arguments
> > > > > >
> > > > > > As the first point of clarification, note that Hu and Huang [1] do *not* use path counting arguments like in [2,3]. These authors use a version of Wick's theorem instead. Indeed, they write in Section 6 that “many previous works adopt path-based approach to study neural network with ReLU activation… Compared with this sum-over-path method, our method is more intuitive and reveals more properties of CK and NTK”. So, these authors are explicitly distancing themselves from path-based approaches, like path counting. In light of this, [1] is not demonstrating how to adapt path-based approaches to ResNets and so [1] does not appear to be evidence contradicting our statements about the challenges of applying path counting to standard architectures. *Is it possible that your comments about [1] were based on interpreting "path counting" differently?* We hope that this resolves this confusion.
> > > > > >
> > > > > > *Side note*: While we all recognize that we are not required to defend our work against claims of novelty regarding [1], since [1] keeps coming up, we will point out that their argument does not have to deal with hypoactivation. In the quote above, the authors of [1] highlight that the advantage of their techniques is that it is more intuitive and "reveal more properties". In fact, we believe path counting arguments can be used to handle the networks studied in [1]. To reiterate why: the architectures studied by [1] have post-activation skip connections. This is a fundamental change to the architecture. (In particular, during analysis, each ReLU activation can be replaced with an iid Bernoulli random variable, unlike the pre-activation skip residual network architectures that we study). We think [1] is an important study towards understanding deep networks though! However, there is considerable additional difficulty that we have tackled in bridging the divide to standard residual architectures. Our paper studies residual networks where the hypoactivation is an important technical issue, which [1] does not address due to the choice of post-activation skip architecture.
> > > > > >
> > > > > > ## On the intractability of path-counting arguments for residual networks
> > > > > >
> > > > > > To address your question on the intractability of path counting, we wrote a section in our most recent reply entitled **"On Path Counting Arguments under Hypoactivation"**, which was our attempt to explain these difficulties in more detail. This section may have been buried. Would it be possible to take another look at this section? (*We have copied the text below for your convenience.*) In particular, we want to emphasize that even computing the moments of a single path is nontrivial, which is the basis for the proofs in [2,3]. The section we wrote sketches some of the numerous difficulties. We apologize if we did not draw sufficient attention to this text in our response. We hope this answers your question.
> > > > > >
> > > > > > ## Summary
> > > > > >
> > > > > > We know you've asked a few more questions. We're organizing and polishing responses! We will submit a response about Conjecture 5's uniformity in depth and your suggestions regarding Theorem 1 in another response soon. But we wanted to get these issues around path-counting resolved before returning to these other important issues we've been discussing.
> > > > > >
> > > > > > **[1]** Zhengmian Hu, Heng Huang. On the Random Conjugate Kernel and Neural Tangent Kernel. http://proceedings.mlr.press/v139/hu21b/hu21b.pdf
> > > > > >
> > > > > > **[2]** Boris Hanin, Mihai Nica. Products of Many Large Random Matrices and Gradients in Deep Neural Networks. https://arxiv.org/abs/1812.05994
> > > > > >
> > > > > > **[3]** Boris Hanin, Mihai Nica. Finite Depth and Width Corrections to the Neural Tangent Kernel. https://arxiv.org/abs/1909.05989
> > > > > >
> > > > > >
> > > > > > # Copy of "On Path Counting Arguments under Hypoactivation" from previous response
> > > > > >
> > > > > > The reviewer asked for a clarification on why the path counting arguments of [1] cannot handle ResNets. The key simplification [1] relies heavily on is replacing each layer’s ReLU activations with an independent diagonal random matrix, where the entries are Bernoulli random variables. However, in the ResNet setting, these diagonal matrices will no longer be independent from each other. If we continue to study the path decomposition using the same approach, we will find that even computing the moments of a single path is non-trivial due to the dependence structure in the Bernoulli random variables.
> > > > > >
> > > > > > For additional intuition, we can consider the distribution of the Bernoulli random variable $\xi := 1_{ z^{\ell+1}_i > 0 }$ conditioned on the previous layer $z^\ell$. Observe that whenever $z^\ell_i \neq 0$, we have the Bernoulli parameter $p := \mathbb{P}(\xi = 1)$ will not be $1/2$. Furthermore, $p$ will depend on the value of the entire vector $z^\ell$. This implies that to compute the moments of a single path, one would need to analyze the dependence structure on neurons *outside of the path*. Consequently, even the most basic component of a path counting argument would fail here.
> > > > > >
> > > > > > We would also like to emphasize that path counting does not work around Conjecture 5. To continue under this approach, one would need to resolve an equivalent technical roadblock caused by hypoactivation.

---

> > > > > > > ### Comment · Reviewer_Kh9k · 2021-08-31
> > > > > > > **Quick response**
> > > > > > >
> > > > > > > I still need to be confirmed the path counting argument can't be used here but  I don't have time to that. I tried to discuss with other reviewers about their opinion about the assumption 5 and the related techniques but there is still no reply. For the decision deadline is approaching, I decide to not to penalize this point and changed my score back to 5.
> > > > > > >
> > > > > > > My main concern now is still assumption 5 is a uniform in depth manner which is too strong to be assumed. The author seems not replied to this point yet.
> > > > > > >
> > > > > > > At this point I think the paper is still not ready for publication, but I still believe the paper is worth be accepted in later conference even with concurrent similar results published recently for this approach is clean and easy. (once assumption 5 is solved or provided the analysis the off-diag terms)

---

> > > > > > > > ### Author Response · Authors · 2021-08-31
> > > > > > > > **See response**
> > > > > > > >
> > > > > > > > We've realized that you've likely missed our response, which we posted as a comment/response to your original review, not as a comment/response to this deep thread, which you cannot even see from the main page.
> > > > > > > >
> > > > > > > > So you can find our response regarding these points just after your original review. You can find it by going to our paper's top level page (or clicking 'show all responses'), finding your original review, and looking just after it... our new response is the first one after your original review). We also attempted to use that response to summarize our discussions somewhat, since we believe we've made some progress along the way.

---

> ### Author Response · Authors · 2021-08-29
> **A New Response Thread to Reviewer Kh9k**
>
> We wanted to start a fresh thread to summarize our discussion with you and attempt to close out some of the multiple threads that have opened up. OpenReview's interface doesn't seem to handle a conversation with as many nested layers as ours!
>
> In the initial review, you write: "the assumption in the paper and conclusion doesn't convince me the new characterization is right and essential." Your review then raises a number of specific concerns. At this point, we have responded to each of these specific concerns. We want to summarize those arguments here, both for you and for other reviewers who might be daunted by the voluminous threads below.
>
> In summary, we feel that every one of your concerns has now been addressed. Your concerns represent opportunities for us to improve our paper with additional discussion and context: we intend to use our extra page in the camera ready to import the essential aspects of our conversation. Indeed, we feel that the concerns you have raised would have been shared by some future readers, and so we are pleased to know that our paper will have improved through peer review.
>
> ## Brief summary
>
> Just to set the stage, in this work, we have introduced a new technique for studying residual networks in the infinite depth and width limit. The approach is motivated by our difficulties adapting path-counting arguments to standard (pre-activation) residual network architecture. The roadblock is the fact that residual networks suffer from "hypoactivation", a phenomenon we are the first to highlight in the context of infinite limits, whereby the inner layers are NOT uniformly distributed, unlike in fully connected architectures. Our new technique for studying the infinite-depth-and-width limit requires that the deviation from uniformity has relative error $O(1/n)$, uniformly in depth. (We credit you for noticing that our conjecture obscured this uniform in depth aspect.) We conjecture that this $O(1/n)$-relative-deviation assumption holds, based in part on some very strong empirical evidence for the conjecture which we regrettably buried in the supplementary. (We agree that we should bring this forward and discuss the uniformity in depth aspect in the main body.) We present a rigorous limit theorem for the distribution of a single input, assuming the conjecture holds. (We appreciate reading your own suggestions for proving the conjecture.)
>
> Our new techniques also readily apply to a slight modification of the standard architecture, whereby we flip each ReLU with probability $1/2$. (This is such a natural modification to the architecture that one might imagine a world in which this was the standard. Preliminary experiments suggest there is no ill effect from this change, unlike moving to, say, post-activation skip connections, which at least requires hyperparameter re-tuning.) We prove that these "balanced" ResNets do not suffer from hypoactivation and establish an (unconditional) theorem regarding limits of these architectures.
>
> ## Closing open threads in our discussion
>
> When we look over our long conversation, there seem to be three open threads to close. Picking up the discussion where we left off, we would like to work out any remaining concerns you might have. In summary:
>
> **1. Re: Pathcounting:** In a recent thread, you raised a new concern: that our claim (that "hypoactivation poses difficulties for path-counting arguments" was undermined by a new ICML paper (appearing for the first time only after the NeurIPS deadline) proving a seemingly similar infinite-depth-and-width limit to ours. In that thread, we have highlighted the fact that this new ICML paper studies a *nonstandard* residual network architecture with post-activation skip connections, and we note that this type of architecture does not have hypoactivation. We have outlined that hypoactivation introduces statistical dependence that prevented us from even computing moments for a single path, nevermind computing moments for multiple overlapping paths. (The ICML authors try to convince readers explicitly that their technique is not path counting. That said, they do discuss "sums over paths" and so it is not clear to us whether their approach would get around the same roadblocks we ran into with moments. Note that this paper appeared after the NeurIPS deadline: we don't believe we should have to compare our work to it. There's also something to be said for having a diversity of approaches.)
>
> On the other hand, this discussion has provided us the opportunity to reiterate several of our key contributions: identifying hypoactivation in standard ResNets; developing a technique to work around it based on the assumption that hypoactivation introduces only $O(1/n)$-relative deviation from uniformity, etc. Can you confirm that we have settled your concerns about our claims about adapting path-counting techniques? Otherwise, we would appreciate it if you could raise specific concerns that we can address head on.
>
> **2. Re: Conjecture:** In your initial review, you raised a concern regarding the plausibility of our conjecture that only $O(1/n)$-relative-deviation from uniformity is caused by hypoactivation, although you have now also suggested several interesting ideas to prove the conjecture. One astute technical concern of yours was whether the error converges uniformly across layers. You are absolutely correct that our conjecture, as stated, does not make it clear that we need the error to be controlled uniformly in depth. We will fix this and add a short discussion. The empirical evidence that hypoactivation converges exponentially fast and thus gives us our desired uniform convergence appeared at the end of the appendix (Figure 9, supplementary). We will mention these results. Finally, we've spent a bit more time thinking about your own proof ideas for the uniform conjecture and we think they are indeed promising. (We encourage you to write them up and solve our open problem.) Can you clarify if we have settled your concerns regarding the plausibility of our conjecture, or otherwise raise specific concerns that we can address head on?
>
> **3. Re: Normalizing Constant:** Finally, one of your initial concerns related to the normalizing constant in Theorem 1. Initially, you raised concerns about rigor. It seems that these concerns about rigor have morphed into a suggestion to simply define $G$ explicitly in the theorem statement. We believe everyone agrees that $G$ (as defined on line 244) is the variable that converges. Note that the theorem statement has always been about convergence of $G$, but we agree it is clearer to just state $G$ explicitly in the theorem statement rather than relate $G$ to $z^\text{out}$, especially because $z^\text{out}$ is diverging potentially. Can you clarify that moving the definition of $G$ into the theorem statement (perhaps annotating $G$ with subscripts for $n, d$ too) will resolve the problems you saw with clarity here? If not, if you could raise a specific concern, we will attempt to address it head on.
>
> In summary, we believe we have resolved all outstanding concerns you have raised in your review. We appreciate your patience during this process and commend you for your level of engagement. Our hope is that you will reconsider your Reject recommendation, as the issues that supported this recommendation are now all addressed. If you disagree, we hope that you will raise specific concerns that we can then address. We believe our work is extremely timely and, despite the flurry of recent activity in this area, presents novel and important results. If the publication of our work is delayed unnecessarily, it will not only delay us from moving on to more challenging problems, but may also have professional ramifications.
>
> We believe our paper makes significant technical contributions to the studying of standard (pre-activation-skip) ResNet architectures, in part by providing a thorough empirical analysis of the “hypoactivation” phenomenon. If you have any further advice on how to reshape the results / discussion to improve the paper, we are happy to make further minor modifications.

---

### Official Review · Reviewer_xrqc · 2021-07-10

**Rating:** 6
**Confidence:** 4

**Summary:**

The paper calculates the forward pass at initialization of resnets in the joint proportional limit of infinite width and depth. In particular, the paper shows that this limit exhibits log-Gaussian behaviors, extending what was previously known for feedforward networks. From here, the paper makes several interesting observations regarding interlayer correlation, enlarged output variance and how to alleviate them. The proof crucially exploits homogeneity of the ReLU activation function, together with an assumed conjecture on the pre-activation vector’s direction being almost uniformly distributed.

**Limitations And Societal Impact:**

Yes

**Main Review:**

There have been a lot of interests in analyses of infinite-width networks, but it is possible — such as in NTK limit — that the infinite width limit results in undesirable properties. One escape from this scenario is to consider the joint limit of width and depth (which was possibly first initiated by Hanin, to the best of my knowledge). This paper is a nice and timely contribution to this line of works; the fact that the limiting behavior is log-Gaussian, which is hypothetically more realistic, is interesting.

I have not spotted any technical flaws yet, and the presentation is nice and clean. A criticism, if any, is perhaps on the implication of the result: would correlated neurons at initialization and somewhat bigger (but not exploding) output variance be a bad thing to have? Would eliminating them via the balanced scheme necessarily be a better thing? I’m not yet convinced by the plots in Appendix C, but to be fair, this is certainly beyond the scope of the paper. Another criticism is perhaps on the limitation of the theoretical result itself, which could have been more informative if the backward pass at initialization could be calculated. I do appreciate the efforts to do this kind of exact calculations though, given that this is typically a challenging calculation to do.

I have two specific remarks:

(1) The proof idea in Section 5 is clean, but it lacks an explanation why the case $\alpha\neq 0$ is much harder than $\alpha=0$ and if there is any new technical element to deal with the complication of $\alpha\neq 0$. In particular, a more detailed comparison with [22] in terms of technicality would be useful, given that this is a purely theoretical paper.

(2) It seems homogeneity of ReLU is critical to obtain log-Gaussian behavior. My impression is that, following the argument in Section 5,

$$
\log \Vert z^{\ell+1} \Vert = \log \bigg\Vert \alpha e_1 + \lambda \sqrt{2/n} \Vert z^\ell\Vert^{-1} \Big\Vert\varphi(\Vert z^\ell\Vert \hat{z}^\ell)\Big\Vert g^{\ell+1} \bigg\Vert + \log \Vert z^\ell \Vert
$$

and as such, when $\Vert z^\ell\Vert$ is not cancelled out inside the first term (e.g. when $\varphi$ is not homogeneous), the terms do not seem to be weakly dependent. It is then unclear why log-Gaussianity should arise. Would it be possible to comment more on this? I somehow feel this is important, since the title of the paper conveys the sentiment that log-Gaussianity is expected to be universal.

A minor comment: In line 42, “linear model” —> “linear method”.

**Time Spent Reviewing:**

6

---

> ### Author Response · Authors · 2021-08-10
> **Response to Reviewer xrqc**
>
> # Summary
>
> We thank the reviewer for the generous comments. In particular, we appreciate the reviewer recognizing the significance of studying the infinite-depth-and-width limit, and the difficulty of these calculations. The reviewer raised several minor but insightful questions, which we will address carefully below. We hope our response will satisfy the reviewer, and contribute towards an improved score to help this paper get accepted.
>
> # On the Effects of Variance and Correlation
>
> The reviewer asked if somewhat larger (but not exploding) variance and correlation at initialization is necessarily a negative property. Indeed, our experiments can only suggest that the balanced ResNet modification does not perform worse. Our main intention in Section 3 was to contrast against the properties of the infinite-width limit, which significantly underestimates output variance and predicts independent neurons. As the reviewer mentioned, the performance of balanced ResNet is somewhat outside the scope of our paper, which is why we did not pursue exhaustive experimentation towards this direction.
>
> # On Backward Pass at Initialization
>
> The reviewer commented that it would be desirable if we could compute the backward pass at the initialization as well. Indeed, as the reviewer mentioned following this comment, this calculation is more challenging due to the dependence structure. Here we will briefly sketch the technical roadblock.
>
> In Section B.6, we have some direct Corollaries regarding the input-output derivative. To compute the derivative with respect to a specific weight, we can recover an input-output type derivative via chain rule with respect to the layer before, more precisely
>
> $$ \frac{\partial z^{\text{out}} }{ \partial W^\ell_{ij} }
> 	= \frac{ \partial z^{\text{out}} }{ \partial z^{\ell} }
> 	\frac{ \partial z^{\ell} }{ \partial W^\ell_{ij} }
> 	= \frac{ \partial z^{\text{out}} }{ \partial z^{\ell} }
> 	\left[ \lambda \sqrt{ \frac{2}{n} } E_{ij} \varphi_+( z^{\ell-1} ) \right] ,
> $$
>
> where $E_{ij} \in \mathbb{R}^{n\times n}$ is all zero except at the $i,j$-th entry. Here, the difficulty arises due to the dependence between $\frac{ \partial z^{\text{out}} }{ \partial z^{\ell} }$ and $\varphi_+( z^{\ell-1} )$. Unlike studying the output alone, this object “skipped” the $\ell$-th layer weight matrix $W^\ell$, which no longer allows the trick in eqn. (16) to rewrite the matrix vector product as a Gaussian vector. However, it’s possible that with a more careful analysis, one can resolve this issue and compute the backward pass as well. We hope our framework will lead towards these future results.
>
> # Comparison With the Feedforward Case
>
> The reviewer asked for some further clarification on the $\alpha \neq 0$ case and a comparison with the work of Hanin and Nica [1]. Indeed, a major simplification in the $\alpha=0$ (i.e. feedforward) case is the symmetry of each layer vector $z^\ell$, in particular, the unit vector $\hat{z}^\ell$ is always uniformly distributed on the sphere $\mathbb{S}^{n-1}$. This allows [1] to replace each instance of ReLU with a multiplication by an i.i.d. Bernoulli random variable, while preserving the output distribution.
>
> In the $\alpha \neq 0$ (i.e. ResNet) case, since the neurons are hypoactivated, i.e. ReLUs are activated less than half of the time, this implies the direction $\hat{z}^\ell$ is no longer uniformly distributed. Consequently, the approach of [1] can no longer write the output as a product of independent random matrices, and therefore rendering the combinatorics part of the path counting argument intractable.
>
> To avoid this problem, we introduced a new approach by analyzing the layer norms and computing their covariances. In fact, this drastically simplifies the argument even in the feedforward case, avoiding combinatorics, moment calculations, and the central limit theorem can be applied explicitly (as opposed to a martingale CLT). Finally, this approach allows us to handle the effects of hypoactivation and interlayer correlations, as they are part of the mean and covariance calculation of layer norms.
>
> # On Non-Homogeneous Activations and Log-Gaussianity
>
> The reviewer observed that a key part of our proof is leveraging the homogeneity of ReLU activations, that is $\varphi_+( cz ) = c \varphi_+(z)$ for all constants $c>0$, and asked if log-Gaussianity would arise for non-homogeneous activations. While we don’t have a complete proof, we can briefly sketch the intuition here. For simplicity, let us consider a feedforward network
>
> $$ z^{\ell+1} = \sqrt{ \frac{c}{n} } W^{\ell+1} \varphi( z^\ell ) ,
> $$
>
> where $c>0$ is the appropriate Kaiming He initialization constant for forward propagation [2], and $\varphi$ is a general activation function. Then using the same approach as eqn.(16) we can write
>
> $$ \frac{ \| z^{\ell+1} \|^2 }{ \| z^\ell \|^2 }
> = \frac{ c \| W^{\ell+1} \varphi(z^\ell) \|^2 }{ n \| z^\ell \|^2 }
> \overset{d}{=} \frac{ c \| \varphi(z^\ell) \|^2 }{ n \| z^\ell \|^2 }  \| g^\ell \|^2 ,
> $$
> where $g^\ell$ is an independent Gaussian vector, and this leads to a similar correlated CLT type sum
>
> $$ \log \| z^{\text{out}} \|^2
> \overset{d}{=} \sum_{\ell=1}^d \frac{ c \| \varphi(z^\ell) \|^2 }{ n \| z^\ell \|^2 }  \| g^\ell \|^2 .
> $$
>
> To reach a log-Gaussian type result, it remains to characterize the correlation in the sequence. While this may be a non-trivial calculation, the intuition here strongly suggests a CLT result is possible, which would lead to a log-Gaussian output distribution.
>
> # References
>
> [1] https://arxiv.org/abs/1812.05994
>
> [2] https://arxiv.org/abs/1502.01852

---

> ### Author Response · Authors · 2021-08-23
> **Response to Reviewer xrqc**
>
> We would enjoy the opportunity to interact with you and answer any questions you might have after reacting to our response. Please let us know if you have any questions.

---

> ### Author Response · Authors · 2021-09-06
> **Any chance to interact?**
>
> Dear Reviewer,
> We've addressed your review, but have not heard back from you as to whether you feel we have addressed your concerns. Whether you update your score may be critical to our paper's acceptance. We hope you'll engage in these last few days.
> Regards,
> Authors.

---

> > ### Comment · Reviewer_xrqc · 2021-09-06
> > **reply**
> >
> > Apologies for the very late reply. I somehow mistakenly thought that I had updated my review.
> >
> > I have read the responses to my review and other reviewers', and many thanks for spending much effort in doing so. You have addressed my questions. Let me share my thoughts with regards to the main points that the reviewers raised.
> >
> > Regarding the conjecture, while I think it would be a solid paper if the conjecture is resolved, I do not think it is a fair point to penalize the paper for not resolving it, unless a reviewer can write explicitly a short and elementary proof -- which I think is the standard practice in maths. Other reviewers had some ideas and discussions on a resolution of this conjecture, and while some are interesting, it's not immediate if a proof can be worked out.
> >
> > On the other hand, the paper would have been much longer had it resolved the conjecture, but NeurIPS readers might not be entirely interested in that. An applied math journal would be a better venue in that case.
> >
> > So it comes down to whether the paper can send new messages or have interesting implications in some way. Log-Gaussianity is interesting, but not entirely new. Hypoactivation is clearly something to take home with (in comparison with previous/concurrent works). It is not yet clear from the paper if this will go a long way.
> >
> > Though I can't say this is a very exciting progress, I envision that there will be some good discussions on the technicality, how to deal with dependency, and how to extend the technique to interesting setups and make stronger statements on the practice, should the paper be accepted. So I remain on the score of 6.

---

> > > ### Author Response · Authors · 2021-09-06
> > > **Response**
> > >
> > > Thank you for the timely and constructive response! We are glad to have the opportunity to engage with you while the window is still open.
> > >
> > > > Regarding the conjecture, while I think it would be a solid paper if the conjecture is resolved, I do not think it is a fair point to penalize the paper for not resolving it
> > >
> > > We agree, of course. However, we would like to add that, through our discussions with the other reviewers, we are now more convinced that formulating the conjecture is one of our main contributions. The utility of the conjecture is demonstrated in part by our demonstration of how it allows us to resolve the exact limiting log-Gaussian distribution. We get the sense that you and several of the other reviewers already agree that our empirical work provides some convincing evidence that the conjecture is likely true.
> > >
> > > > So it comes down to whether the paper can send new messages or have interesting implications in some way. Log-Gaussianity is interesting, but not entirely new. Hypoactivation is clearly something to take home with (in comparison with previous/concurrent works). It is not yet clear from the paper if this will go a long way.
> > >
> > > We'd like to respectfully push back on this comment.
> > >
> > > To begin, we'd like to suggest that demonstrating log-Gaussianity is not our chief contribution. We now understand that our chief contributions are 1) recognizing and defining hypoactivation, 2) measuring its size is not too large empirically, 3) formalizing a conjecture around this, 4) exploiting this conjecture to exactly determine the output distribution of the network on an input and 5) demonstrating through empirical work that our predictions are remarkably accurate, much more so than NNGP ones. Really #4 and #5 serve to show that our conjecture has a lot of power.
> > >
> > > We've named the paper "the future is log-Gaussian" but perhaps, in light of the above, we should have highlighted hypoactivation instead. (It's not too late to tweak the title, with your input, so this isn't really a reason to not recommend acceptance!) So, in summary, regarding "Log-Gaussianity is interesting, but not entirely new", we would say: the limiting output distribution is what it is. It has a log-Gaussian component. But, the key point of our paper is making the first analysis of the standard ResNet architecture.
> > >
> > > Of course, you are referring to the fact that log-Gaussianity has been shown in fully connected feedforward networks. Regardless, ours is the *first* paper demonstrating that it arises theoretically in a very distinct architecture (ResNets) and verifying our precise calculations (not only the log-Gaussianity but precisely its parametrization etc). Note that we are not currently responsible for work appearing simultaneously. (In this case we are referring to ICML paper.) *However, should our work be rejected, the next set of reviewers will say "you were not even the first to demonstrate log Gaussianity in residual networks theoretically."* It seems rejection has steeper costs in these fast moving areas.
> > >
> > > We're not quite sure what it would mean "if [hypoactivation?] will go a long way". In a technical sense, hypoactivation is a roadblock that *every* researcher will have to contend with. It's again a material fact of ResNets that we have uncovered. Our work is extremely timely---now is the time for the community to be made aware of this phenomenon. Our conjecture unleashes researchers to keep pressing forwards. Our work demonstrates that precise and accurate calculations can be made using the conjecture. Our conjecture will now be a target for ambitious theoreticians. So, we would argue, hypoactivation will surely live on forever. (If we've misunderstood you, we'd appreciate some clarification.)
> > >
> > > > Though I can't say this is a very exciting progress, I envision that there will be some good discussions on the technicality, how to deal with dependency, and how to extend the technique to interesting setups and make stronger statements on the practice, should the paper be accepted. So I remain on the score of 6.
> > >
> > > Of course, we're very disappointed to read this. Could you offer any concrete suggestions on how we would have had to have changed the paper, assuming we both agree that the conjecture need not be resolved for the work to be published? More results? The paper is already way too long. We have identified a critical new phenomenon (hypoactivation) that every researcher working on this problem must grapple with. We've measured this phenomenon empirically to understand the scale of it, formalized this into a conjecture we believe is true, and then marched on to complete the first analysis of a standard ResNet’s initial distribution, verifying the final predictions empirically.
> > >
> > > We'd like to convince you to argue for acceptance. Here are a few arguments:
> > >
> > > 1. At present, given the scores, the decision will be in the hands of the AC. But reading your own final paragraph, it sounds like you believe that our paper will lead to a host of useful activities, including "discussions on the technicality, how to deal with dependency, and how to extend the technique to interesting setups and make stronger statements on the practice". Many NeurIPS papers don't even reach this level. You are clearly not recommending rejection, but that may well be the outcome given the scores. Would it not be disappointing for this work to be rejected?
> > >
> > > 2. Indeed, our work is timely *now*. If our work is needlessly delayed by 4-9 months, some of the results we have shown will start to appear in similar forms: a good chunk of our novelty will be out the window. Researchers will read our ideas and start to absorb them into their own work. Indeed, this area is moving so rapidly that our paper may appear less timely in the future. (Thought experiment: what if our conjecture is resolved by other authors in the interim. We may never be able to publish our work.)
> > >
> > > 3. You may imagine small improvements/extensions that could be made, but are these really a reason to hold up this work? We believe it made sense to publish this result now and in this form because we have identified key phenomena and shown the way forward via our conjecture and subsequent analysis. There is always more work that can be done. As we said to another reviewer, no paper is perfect. And in this case we believe the saying "perfect is the enemy of the good" is very pertinent. A straight Accept rating doesn't say this paper is perfect, it says it should be accepted and it makes this more likely. We hope you don't mind us pushing back somewhat to argue that we've made an important and timely contribution. We'd prefer not to leave our fate to the wind.
> > >
> > > We appreciate your consideration.
> > >
> > > Regards,
> > >
> > > Authors

---

### Official Review · Reviewer_QBeZ · 2021-07-13

**Rating:** 8
**Confidence:** 3

**Summary:**

The papers studies the infinite width/depth limit of ReLU ResNets, to contrast it to the more usual infinite width/fixed depth limit. They show (under the assumption that a certain conjecture is true) that in the infinite width/depth limit, the limiting distribution of the outputs of the network at initialization (for i.i.d. Gaussian entries) is not Gaussian but log-Gaussian. While this was known for fully-connected networks, proving it for ResNets is more difficult due to dependence between the layers. This is related to the fact that less than half the neurons are active and that there exists correlations between layers of the network (which does not happen for fully-connected networks). These two properties of ResNets however lead to very large variances at initialization which can make training difficult, the authors therefore propose a balanced ResNets (where the activation at each neuron is chosen randomly between the ReLU and the "negative" ReLU) which ensures that half the neurons are active and removes the correlations, however the variance at initialization (though smaller than that of ResNets) can still be large (much larger than what is predicted by the infinite width limit). Finally the results are checked empirically, showing that even for small depth/width ratios (say 0.1) the combined with/depth limit describes the statistics of DNNs more accurately than the fixed depth, infinite with limit.

**Limitations And Societal Impact:**

The requirements of a Conjecture for the proofs of the main Theorem is of course a weakness, though the Conjecture itself is well discussed and checked empirically. This work is theoretical and has no direct societal impact.

**Main Review:**

As discussed by the authors in the introduction, it is important to understand the limits of the NTK regime. While a lot of recent work has focused on the active regime for finite depths (especially for shallow networks with the mean-field limit) fewer papers have studied the large depth limit. The fact that less than half of the activations are active at initialization and the correlations between layers was unexpected to me, but the paper does a good job explaining how these arise, and show the impact these quantities have on the distribution of the outputs. The numerical experiments show the relevance of this combined limit even for small depth/width ratios, yielding much better approximations of some statistics of DNNs than the infinite width limit.

The article is well written, the Conjecture is well discussed and the sketch of proof is very helpful. I am not familiar enough with this line of work to determine whether the Conjecture is reasonable.

**Time Spent Reviewing:**

3

---

> ### Author Response · Authors · 2021-08-10
> **Response to Reviewer QBeZ**
>
> We thank the reviewer for the generous comments and score. The reviewer has clearly recognized the significance of our work in the context of existing literature, and we are glad to see the reviewer did not struggle with any issues of clarity while reading. We were also surprised by the observation that less than half of the neurons were active, which is why we considered formulating Conjecture 5 as a novel contribution. We hope the reviewer enjoyed reading our work, and that our work gets accepted here at NeurIPS.

---

### Official Review · Reviewer_SxtD · 2021-07-15

**Rating:** 5
**Confidence:** 4

**Summary:**

The authors study signal propagation in randomly-initialized feedforward
residual neural networks with gaussian weights, with architectures consisting
of one fully-connected layer per each residual block in the network.
Conditional on a certain conjecture about distributions of normalized
preactivations in the residual network, they prove theorems that precisely
articulate the distribution of the norm of the network output in a limit where
the network depth $d$ and width $n$ simultaneously tend to infinity and are
directly related, and as a byproduct articulate precise dependences of the mean
and variance of this output, as well as the inter-layer correlations of the
network, as functions of the depth and width. The authors claim interesting
consequences of this conditional analysis relative to predictions one would get
by considering a limit where just the width $n$ is taken to be infinite (and
then possibly the depth) -- for example, they show that standard ResNets have
less than half of their neurons in each layer activated, which can lead to
certain unstable output behaviors, and that networks in this limit (in
particular when $d > n$ as the limit is being taken) may have
exponentially-large output variances, which may be challenging to mitigate
(Section 3). The authors justify the conjecture underlying their theoretical
results with Monte Carlo simulations and a heuristic argument.


**Limitations And Societal Impact:**

(checklist 1a) The theoretical results in section 2 onwards are completely
clear, and accurately present all limitations of the results, but I do not
think the paper's contributions as framed in the abstract and introduction
accurately reflect the paper's theoretical results' dependence on the unproven
Conjecture 5. E.g. line 9 "We show theoretically ..." in the abstract should
have something like "Assuming a certain conjecture on the distributions of the
preactivations in the network, ..." prepended to accurately reflect what the
paper shows; lines 58-60 in the introduction do not unambiguously state that
"find[ing] a simple approximation" is an empirical/conjectural activity, rather
than theoretical"; line 89 second bullet in relation to the previous bullet is
similar.



**Main Review:**

## Summary (wrt rating)

The paper's study of signal propagation properties in simple ResNet
architectures is timely, and the conclusions are of potential general interest:
the authors demonstrate that infinite-width regime predictions about signal
propagation in residual nets may not accurately reflect the performance of real
networks, given certain depth dependences that vanish when $n \to \infty$
completely (eqn below line 187 is especially unsavory). However, the paper has
several limitations that lead me to rate it as such:
- Almost all novel theoretical results in the paper hold only conditionally on
  a certain conjecture about preactivation distributions in the network, which
  the authors justify intuitively (section 4) but do not prove; this conjecture
  seems to encapsulate most of the technical difficulty unique to the analysis
  of ResNet signal propagation relative to the fully-connected case, and it
  therefore does not seem appropriate to consider the work a contribution to
  the literature on the theoretical analysis of ResNets in this state.
- In light of the previous point making most theoretical results in the paper
  only conjectural, I would expect a stronger practical component to the work
  in order for it to be suitable for presentation at NeurIPS; however, most
  theoretical results in the paper are limited to experiments verifying the
  theoretical predictions (for ResNets with one-layer blocks) using Monte Carlo
  simulation, rather than verifications on practical architectures or
  demonstrations of predictions relevant to practice and mitigation strategies
  (e.g. in this literature connections to the "vanishing-exploding gradient
  problem" are often made; here the discussion in Section 3 seems mostly to
  survey the theory's predictions rather than interpret it for practice).
- The contributions do not seem to be sufficiently contextualized in the
  literature.

It seems to me that the quality of the work would be greatly enhanced by either
a proof of Conjecture 5 (which would make it acceptable for the work to have a
minimal practical component), or an expansion of the practical component of the
work (as above). I hope the authors will correct me if I have unfairly
interpreted the theory, its dependence on Conjecture 5, and its implications.

## Detailed Comments
I will intermix both strengths and weaknesses in the subsequent discussion.

### Theoretical aspects
- The mathematical writing of the authors in sections 2 onward is (rather
  refreshingly) clear and rigorous, making it painless to understand and
  interpret their theoretical results.
- It seems to be a severe limitation of the work (as above) that almost all
  novel theoretical conclusions in the paper depend on the unproven Conjecture
  5 (as I see it -- the characterization of the moments of the random variable
  $G$ and its asymptotic behavior via a CLT in Theorem 1; the precise
  calculations of certain architecture-dependent constants appearing in these
  moments in Proposition 2). This effectively makes the theoretical results
  conjectural, only supported by Monte Carlo simulations -- and I am not sure
  it is possible for such simulations to be adequately convincing at this
  level, given that the state space one needs to explore is a two-parameter one
  in (d, n). Although I suppose it could be fine in certain cases to establish
  and present theorems that depend on a conjecture (e.g. conditional hardness
  results in TCS; implications of the Riemann hypothesis...), the nature of the
  conjecture in this paper (a statement about the distribution of the norms of
  the normalized preactivations across layers of the ResNet) seems to me to
  encapsulate most of the novel difficulty (i.e., what's different from the
  well-understood FFNN case) in the analysis of things like the
  network output norm and the inter-layer correlations in the ResNet context,
  to the extent that it feels more like the present work skips key aspects of
  the analysis of the problem compared to these examples of conjectural results
  in other fields. I hope the authors will correct me if I have been unfair in
  the way I have construed this issue.
- The assertions for "Balanced ResNets" (Theorem 4), which do not
  require Conjecture 5, seem to be treated by the authors as "trivial" (the
  proof of Theorem 4 in the appendix is described as such) -- I would
  appreciate clarification if there are any novel aspects to these proofs (e.g.
  parts of the proof of Theorem 1 in the appendix that need to be carried over)
  or whether the proofs become essentially identical to the fully-connected
  case with this architectural modification. (The former case would speak to
  the novelty of the authors' work.)
- Question: does the equidistributional assertion in the proof sketch (eqn
  (17)) require some additional justification, given that the random variables
  one is asserting equidistributionality for (the factors in the product) are
  dependent? I imagine this is benign; I just point it out since I didn't find
  an additional justification for this manipulation in the proofs in the
  appendix.

### Practical aspects
- Somewhat related to the next section, it seems like there is a large existing
  literature on initialization schemes for ResNets, and it would be of interest
  to hear the authors discuss any implications of their results relative to
  these practical methods (and/or dismiss them as not directly relevant, etc.
  if that is the case). For example, the authors mention Stable ResNet and the
  work of Hanin and Rolnick in lines 158-163; what is the precise relation to
  these works? (the authors mention previously that Stable ResNet considers
  things in the infinite-width setting, which seems good; the authors explain
  their Theorem 1 as more precise than Hanin and Rolnick's, but it would be
  nice to see in what sense (given as well that Theorem 1 is conjectural)).
  Also, what about comparisons to other initialization approaches that involve
  (as far as I can tell) mitigation strategies for signal propagation issues in
  ResNets -- e.g. [1-3] below.  It seems like it would be nice to have this
  material presented in a Related Work section to easily understand how the
  authors' contribution fits in here (maybe these approaches imagine the widths
  are infinite, for example, giving evidence to the authors' claim that the
  "d/n constant" setting has been neglected).
- Question: why have the authors restricted their study to "vanilla ResNets",
  which here means a standard ResNet where each block just has one fully
  connected layer? It seems in the literature one commonly sees networks
  studied that admit the possibility of multiple fully connected layers per
  block; and moreover that these are common features in practical ResNets (I am
  certainly not an expert here but I think ResNet-50 and so on have this
  feature?).

### Context
The authors should perhaps add a related work section and compare more extensively to
prior art there -- I feel this would greatly benefit the paper by making the
novelty of the authors' contributions clear. I will discuss a few specific axes
here below -- perhaps the authors can clarify if they intentionally omitted
discussion of these works because they see them as not directly relevant.

- The identification of "hypoactivation" as a key driver of vanilla ResNet
  signal propagation seems potentially of broad interest (even without rigorous
  guarantees to characterize it). It seems like this is a novel discovery of
  the authors -- it feels like additional discussion of relationships to other
  ResNet initialization schemes + analyses in some kind of a new related work
  section (as above) would go a long way towards emphasizing the authors' novel
  insight in this connection.
- I find the motivation around infinite depth-and-width limits in the
  introduction to be somewhat incompletely specified. For example, there is a
  rather large body of work that focuses on nonasymptotic guarantees for
  training and generalization of FFNNs in the NTK regime, and many of these
  apply to deep networks -- e.g. the authors cite Allen-Zhu et al., Zou et al.,
  and Du et al. in the first sentence in connection to the large-width regime,
  but all of these works allow one to take certain infinite-depth-and-width
  limits in the theorems they prove and obtain various guarantees in this limit
  (for training, signal propagation, etc.), and these aspects of these results
  seem to be overlooked in the discussion starting at line 35. There also may
  be some relevant omitted works in this connection (e.g. [4-7] below).
  I find this kind of omission to be surprising given some of the later
  motivation, e.g. sentence at line 63 about "finite behaviour".
- The remainder of the paragraph starting at line 35 also seems somewhat
  dubious without additional clarification relative to these works -- e.g. in
  the sentence starting at line 38 (and somewhat in the remainder of the
  paragraph), did the authors mean for this discussion to apply only to
  networks where $d / n$ asymptotically approaches some constant value? This
  would rule out the applicability of many of the aforementioned works,
  although some may remain relevant, as the quoted reference here discusses
  this infinite depth-and-width limit only for diagonal elements of the NTK and
  not guarantees for the entire training process (for example, I believe that
  certain technical lemmas in Allen-Zhu et al. apply to regimes where
  the depth can grow linearly with the width). It might be beneficial to add
  some additional clarification to this discussion.
- The "Balanced ResNets" proposal seems to be a standard 'trick' in the
  theoretical deep learning literature in order to introduce theoretical
  tractability -- for example, I have seen this used prominently (for a
  slightly different purpose) in [8] below, and that work cites other previous
  deep learning theory papers. I would not imagine the authors need to credit
  these works for the idea, as this seems to be even more broadly a standard
  trick in probability theory, but rather to reference relevant prior art so
  that this does not seem to be claimed as a unique architectural innovation.
- It would be nice to have a comparison to [9], which seems relevant --
  although I cannot fault the authors for not including this as I do not
  believe the work was available until after the authors submitted their work.
  Nevertheless, it seems relevant (and possibly an opportunity to emphasize the
  novelty of the authors' perspective if there is a $d/n$ comparison to be
  made). [10] also seems like it should be compared to.


[1] http://arxiv.org/abs/1901.09321

[2] http://proceedings.mlr.press/v119/blumenfeld20a.html

[3] https://arxiv.org/abs/1906.02341

[4] https://arxiv.org/abs/1904.11955

[5] https://openreview.net/forum?id=fgd7we_uZa6

[6] https://openreview.net/forum?id=O-6Pm_d_Q-

[7] https://arxiv.org/abs/2006.06657

[8] https://openreview.net/forum?id=rkllGyBFPH

[9] http://proceedings.mlr.press/v139/hu21b.html

[10] https://arxiv.org/abs/2001.10460

**Time Spent Reviewing:**

6

---

> ### Author Response · Authors · 2021-08-10
> **Response to Reviewer SxtD**
>
> # Summary
>
> We thank the reviewer for the careful and constructive comments. The reviewer clearly understands the context of which this work fits within the literature, and sees the importance of studying the infinite-depth-and-width limit. In particular, we thank the reviewer for the generous comments on the clarity of our mathematical presentation, and that even the identification of “hypoactivation” alone (without rigorous characterization) is novel and potentially of broad interest.
>
> While this is an in-depth review, and therefore hard to summarize, we think at the center of the concerns raised by the reviewer is the following question: how significant are the  theoretical results for the balanced ResNet (Thm. 4) and do they differ from existing results on feedforward networks. In the sections to follow, we will first explain the novelty of our approach and techniques compared to existing results on feedforward networks, which regrettably could have been better presented in the paper. Then, applying these techniques to vanilla ResNets, it will be clear that Conjecture 5 does not “encapsulate most of the technical difficulty,” but rather identifies the final cornerstone required to resolve the problem. Finally, we will discuss insights from the Monte Carlo simulations on hypoactivation, and how these provide strong evidence in support of the conjecture.
>
> We hope the reviewer will be satisfied with our response, and convinced to increase the score, thus helping this paper get accepted.
>
> # On Technical Novelty Beyond Feedforward Networks
>
> The reviewer wanted to understand our theoretical contributions on residual networks beyond the known feedforward results [1]. In particular, we hope to address the concern “this conjecture seems to encapsulate most of the technical difficulty.”
>
> Indeed as the reviewer suspected, if we isolate the proof of Balanced ResNet (Thm. 4) from the main proof of Thm. 1, it already contains a new approach by analyzing layer norms and its variances, which is not straightforward. In contrast to the path counting argument in [1], this approach drastically simplifies the proof. In particular, there are no combinatorics, no moment calculations (for every even moment), and the central limit theorem can be applied explicitly (instead of a martingale CLT). Consequently, we find this proof far easier to interpret.
>
> Furthermore, this approach allows us to compute the effects of interlayer correlation, whereas pathing counting arguments of [1] cannot handle this effect. More specifically, [1] relies heavily on replacing each layer’s ReLU activations with an independent diagonal random matrix, where the entries are Bernoulli random variables. However, in the ResNet setting, these diagonal matrices will no longer be independent from each other, rendering the combinatorics of path counting intractable.
>
> Finally, we arrive at the only problem remaining: the hypoactivation effect shifts the direction of each (unit) layer vector $\hat{z}^\ell$ such that it is not uniform on the sphere $\mathbb{S}^{n-1}$ (as opposed to the feedforward case). Observing the statement of Thm. 1, the only effect hypoactivation has is shifting the mean of $G$, whereas the interlayer correlation effects are already characterized by Prop. 2.
>
> Therefore, Conjecture 5 is not hiding the main technical difficulty, but rather filling in the role of identifying the final problem, to which we have an accurate Monte Carlo verification of (to be discussed next). We further emphasize that the conditional results of Thm. 1 and Prop. 2 are already sufficient to predict the output density, which is then plotted in Figure 1, showing a very accurate match to finite size ResNets.
>
> # Monte Carlo Verification of Conjecture 5
>
> The reviewer raised a concern regarding the experiments not sweeping the depth variable $d$. This won’t be necessary for three reasons:
>
> 1. In verifying the experiment for some value of $d$, the experiment simulates all intermediate layers $1,2,\cdots,d-1$ thereby running through all values of depth up to $d$.
> 2. Based on additional simulations in Figure 9, we found that hypoactivation, as a function of depth, quickly converges to an equilibrium in a way that is analogous to autoregressive processes. In other words, it reaches stationarity exponentially fast. Therefore, varying the value of depth will have minimal effect once the depth is large enough to reach this equilibrium, which happens quite early at approximately $d=20$.
> 3. One can view the network as a discrete stochastic process in $\mathbb{R}^n$, where the layers play the role of time. (So depth $d$ corresponds to time $d$). To verify the conjecture, one only needs to verify that the equilibrium distribution of this stochastic process has observables which are within $O(1/n)$ of that of a Gaussian process.
>
> In summary, verifying Conjecture 5 will only require Monte Carlo experiments varying the values of width, for a fixed sufficiently large depth.
>
> # Other Minor Comments
>
> ## Comparison to Related Work
>
> We thank the reviewer for providing a list of related work. We will add remarks on the relationship with each paper. We want to briefly comment on the point of non-asymptotic guarantees. Despite much of the work in this area having non-asymptotic results, to the best of our knowledge, they all rely, in some way, on the width being much larger than the depth (formally, depth is treated as constant), which makes them not comparable to our work. In particular, a key condition requires the NTK to be approximately deterministic and time constant, which is emphatically not the case in the large depth regime [2]. If the reviewer can provide a precise reference to Allen-Zhu’s technical lemma which scales linearly with depth, we can provide a direct comment on that.
>
> The reviewer also asked us to comment on [3], which was available after the NeurIPS submission deadline. Indeed [3] proved a similar log-Gaussian result to ours, however, we would like to point out that the authors decided to study a ResNet architecture that added skip connections after ReLU activations, which is known to perform worse in practice [4]. This subtle change is the root cause of hypoactivation and interlayer correlations, which [3] did not have to handle. That being said, our balanced ResNet theorems can be immediately tweaked to handle this case.
>
> ## Answer to Specific Questions
>
> The reviewer is correct that the equality in distribution issue for eqn. (17) is benign. There are two steps to this calculation: first, replacing $Wv$ where $W$ is a Gaussian matrix with $||v|| g$ where $g$ is a Gaussian vector (as explained in eqn. (14)) and, second, factoring out an orthogonal transform $O$ out of the norm. Since the weight matrices $W$ are independent from layer to layer, the resulting $g$ vectors will also be.
>
> Regarding multiple fully connected layers within each residual block: Indeed the same argument we presented can be easily extended to this setting. We only need to modify eqn. (16) to contain multiple products of $\| \varphi(\hat{z}) \|$ type terms in front of the Gaussian vector $g$. We would be happy to describe this extension in the appendix, but it adds nothing new technically.
>
> # References
>
> [1] https://arxiv.org/abs/1812.05994
>
> [2] https://arxiv.org/abs/1909.05989
>
> [3] http://proceedings.mlr.press/v139/hu21b.html
>
> [4] https://arxiv.org/abs/1603.05027

---

> > ### Comment · Reviewer_SxtD · 2021-09-01
> > **response**
> >
> > Dear authors,
> >
> > Thanks for your rebuttal and for following up. I would like to discuss some
> > questions about your rebuttal, then summarize my current outlook on the paper
> > and hopefully we can go from there with further discussion.
> >
> > ## Technical Novelty Beyond Feedforward Networks
> >
> > I have to admit a lack of deep familiarity with the path counting argument of
> > Hanin and Nica you allude to, and as a result I found myself unable to fully
> > appreciate your points. Here are a few things that I failed to understand:
> >
> > - Isn't this path counting argument more general than the argument in the
> >   present submission, in that it gives results for diagonal elements of the NTK
> >   as well (i.e. products of weight gradients at the same input)? I can fully
> >   accept that the present approach avoids cumbersome details that come with the
> >   path counting argument, but I would have thought this is due to the present
> >   approach being tailored to a simpler problem, and possibly not generalizable
> >   beyond this problem (if I understand correctly, this approach just looks at
> >   input gradients -- like in Corollary 24 of the submission).
> > - Beyond this, I am not totally clear on what is the precise novelty of the new
> >   approach that you are claiming here. Are you claiming novelty of the idea to
> >   analyze the feature/output norms by writing them as a product of random
> >   variables, taking logarithms to turn this into a sum, Taylor expanding and
> >   calculating means/variances? Or is it wrapped up in the application of a
> >   different non-martingale CLT? I might be missing something, but I am under
> >   the impression that this kind of "linearizing a product" approach to signal
> >   propagation in feedforward networks is well-established: e.g. it can be found
> >   in Lemma 7.1 of [1], (with martingale tools), or Lemma D.2
> >   of [2], (with linearization and Bernstein). Both these
> >   results prove subgaussian-type tail bounds, yielding full moment control (in
> >   particular means and variances). For ReLU FFNNs, these basic approaches can
> >   be generalized fairly systematically to obtain linear-regime NTK control as
> >   well.
> > - I might be missing something, but isn't the interlayer correlation
> >   contribution equal to zero in Theorem 4 (c.f. eqn 3)? I was under the
> >   impression that the computation of these correlations relied on the
> >   conjecture, which is one major reason why I felt it encapsulated some key
> >   difficulties in the problem (this issue seems to completely disappear in the
> >   balanced case!).
> >
> > On the whole, I have very little issue believing that Conjecture 5 is true
> > (modulo some possible small inessential modifications which may always creep in
> > -- for example, does the conjecture seem to hold for arbitrary $\alpha$ and
> > $\lambda$, or only the specific symmetric case shown in Figure 2). My issue is
> > that I feel the present submission is very much a theory paper, and that in
> > this context, I don't think it is appropriate to present a conjecture and only
> > verify it with MC sampling experiments: these are a tool we use to formulate
> > conjectures, but the core contribution in a highly-specialized, technical
> > setting like the present one does not seem to me to be in formulating
> > conjectures of this type, but in proving them. Again, I would find it
> > completely suitable for this to be left as a conjecture if the paper made more
> > connections to practice, but from my reading of the paper and the rebuttal that
> > seems to be not the primary focus here.
> >
> > ## Monte Carlo Verification
> >
> > Thank you for the clarifications here. I understand it isn't possible to
> > present figures in OpenReview, so I will not pry too much here, but I would
> > strongly urge that if the authors plan to pursue submission of the work in its
> > present state, they expand the discussion of the conjecture to include some of
> > the evidence discussed in points 2 and 3. It seems to me like this would be
> > useful to any readers who are interested in trying to build off the authors'
> > work. Perhaps this could be added to the supplementary.
> >
> > I would also tend to feel like additional technical discussion of possible
> > approaches and pitfalls to establishing the conjecture would be very much
> > merited in Section 4 of the submission. My current reading of Section 4 is that
> > it very nicely presents several heuristics that make the (essential) truth of
> > the conjecture believable, but it does not give any intuition about why this
> > conjecture is hard to prove, what types of approaches will not work, and what
> > types of approaches could. (For this reason, I sort of assumed the authors were
> > actively working on it themselves.) Indeed, I found myself unable to resist
> > trying to mount an attack on the conjecture: I am curious whether it is clear
> > to the authors that a relatively-simple inductive approach might not be a
> > viable approach here. For example, we have a formula (from homogeneity)
> > $$ \mathbb{E} [ \\| \varphi_+ ( \hat{z}^{\ell} ) \\|^2 ]
> > = \mathbb{E} \left[
> >   \frac{
> >   \\| \varphi_+( \alpha \hat{z}^{\ell-1} + \lambda g^{\ell} \\| \varphi_+(
> >   \hat{z}^{\ell-1} ) \\| ) \\|^2
> >   }
> >   {
> >   \\| \alpha \hat{z}^{\ell-1} + \lambda g^{\ell} \\| \varphi_+(
> >   \hat{z}^{\ell-1} ) \\| \\|^2
> >   }
> > \right],
> > $$
> > which seems like it could be amenable to induction. I can see that these
> > calculations quickly become technical, and that it seems to be necessary to
> > simultaneously perform induction on the mean and variance quantities (which
> > seems like somewhat of a can of worms), but it is not clear to me that this
> > approach is fundamentally not workable. I would appreciate some insight into approaches the authors have tried and possibly find promising here.
> >
> > ## Related Work
> >
> > Thank you for the clarification about the ICML21 paper I linked. Regarding the
> > non-asymptotic results, I do agree with you about the full NTK analyses not
> > being relevant here -- since large changes during training will occur in the
> > NTK in the linear regime -- but since the present submission studies only
> > signal propagation issues, I had in mind some lower-level lemmas in these
> > results that apply in the linear regime. For example, one of the results I had
> > in mind here was the Lemma 7.1 of [1] I mentioned above, and I think there are
> > a few others in section 7 of that paper, like Lemma 7.3; in addition [2] has
> > linear-regime concentration bounds for the NTK (Theorem 2), although there are
> > log factors.  There may be differences in the setting/limits the authors are
> > considering that make these works not directly relevant; at the same time, it
> > seems therefore that it might be appropriate to expand the discussion a bit to
> > highlight these differences.
> >
> > ## Outlook
> >
> > I currently tend to maintain my initial score. I believe the paper and its
> > value to the field will be greatly enhanced by a proof of Conjecture 5 -- in my
> > current understanding, based on my reading of the paper and the rebuttal, this
> > conjecture indeed encapsulates the key statistical and "dynamical" challenges
> > in understanding signal propagation in correctly-parameterized residual
> > networks (for example, the fact that hypoactivation and inter-layer correlation
> > issues are only present in the vanilla ResNet seems to me to be strong evidence
> > for this viewpoint), and a proof of this conjecture will therefore represent a
> > highly novel technical contribution to the study of ResNet signal propagation.
> > Given the questions I wrote above, though, I would appreciate additional
> > discussion if the authors continue to feel the current submission is enough.
> >
> > ## References
> >
> > [1] http://arxiv.org/abs/1811.03962
> >
> > [2] http://arxiv.org/abs/2008.11245

---

> > > ### Author Response · Authors · 2021-09-01
> > > **Response 2/2**
> > >
> > > Response 2/2
> > >
> > > # On Path Counting
> > >
> > > We do not believe that path counting is a “more general technique”; despite significant efforts personally spent attempting to apply path counting to this problem, we have found fundamental obstructions when applied to the vanilla ResNet architecture, due to hypoactivation, whose discovery is a central contribution of this work. Every researcher working on standard ResNet architectures must be aware of hypoactivation and our work provides some critical guidance to researchers working on these bleeding edge questions.
> > >
> > > We have copied a section below from a response to another reviewer explaining why we think path counting won’t work here. In particular, we do not see how the hypoactivation phenomenon could be analyzed from the path counting technique without significant changes/additions to the method.
> > >
> > > # Copy of "On Path Counting Arguments under Hypoactivation" from a response to another reviewer
> > >
> > > You asked for a clarification on why the path counting arguments of Hu and Huang [1] cannot handle ResNets. The key simplification that [1] relies heavily on is replacing each layer’s ReLU activations with an independent diagonal random matrix, where the entries are Bernoulli random variables. However, in the ResNet setting, these diagonal matrices will no longer be independent from each other. If we continue to study the path decomposition using the same approach, we will find that even computing the moments of a single path is non-trivial due to the dependence structure in the Bernoulli random variables.
> > >
> > > For additional intuition, we can consider the distribution of the Bernoulli random variable $\xi := 1_{ z^{\ell+1}_i > 0 }$ conditioned on the previous layer $z^\ell$. Observe that whenever $z^\ell_i \neq 0$, we have the Bernoulli parameter $p := \mathbb{P}(\xi = 1)$ will not be $1/2$. Furthermore, $p$ will depend on the value of the entire vector $z^\ell$. This implies that to compute the moments of a single path, one would need to analyze the dependence structure on neurons *outside of the path*. Consequently, even the most basic component of a path counting argument would fail here.
> > >
> > > We would also like to emphasize that path counting does not work around Conjecture 5. To continue under this approach, one would need to resolve an equivalent technical roadblock caused by hypoactivation.
> > >
> > > **[1]** Zhengmian Hu, Heng Huang. On the Random Conjugate Kernel and Neural Tangent Kernel. http://proceedings.mlr.press/v139/hu21b/hu21b.pdf
> > >
> > > # On the Role of the Conjecture
> > >
> > > We get the sense from reading your response that you are thinking of the conjecture as “assuming the final answer we want” thereby sidestepping the difficulty of the problem. On the contrary, we think of the conjecture as “assuming that the error terms are at the order that one would morally expect them to be”. The role of the conjecture is just ruling out that the error could be, say, $\Omega(1/\sqrt{n})$ (which would then lead to an explosion in the infinite depth and width limit). Once we have established that the error term is the *size* one expects, we go on to exactly calculate (using the $J_2$ function in a novel way) the precise formula for the interlayer correlation! So in other words, the conjecture is the jumping off point to calculate exact formulas. It does not assume any answer itself!
> > >
> > > # Novelty in Our Work
> > >
> > > You are clearly very knowledgeable of the history of the field and intimately aware of the many mathematical pieces that are in the paper. However, by focusing on the novelty of the individual mathematical tools used, we think you are undervaluing the novelty of the paper as a whole. To illustrate what we mean: imagine a researcher who is an expert at feedforward nets attempting to analyze the vanilla ResNet we studied here. After applying the familiar techniques, we believe this research would arrive at the *wrong answer*; their prediction would not account for the added effect of interlayer correlations and hypoactivation because they would not be aware of these effects and would have incorrectly baked the assumption of no hypoactivation into their calculations! (Indeed, this is precisely what happened to us at first: we only discovered the hypoactivation when comparing our initial prediction coming from the feedforward net theory to Monte Carlo simulations). The novelty of our paper in this light is the discovery of the hypoactivation/interlayer-correlation phenomenon, understanding why this happens by boiling it down to its essence in Conjecture 5 (which bounds only the size of the effect), and carefully calculating exactly the effect starting from the bound in the conjecture.
> > >
> > > # Calculating Interlayer Correlations in Thm 4
> > > It is true that, for balanced ResNets, the interlayer correlations are 0 and this does not rely on the conjecture. This fact is precisely *why* we introduced the balanced ResNet architecture; balancing the network precisely sets up Thm 4 to work as written. So the novelty here is to define the balanced ResNet in such a way as to make Thm 4 work but crucially to do so *without* destroying a key property of the ResNet. In particular, skip connections remain preactivation, and so the network maintains the "perturbation of identity" interpretation. Our experiments suggest that, unlike other nonstandard modifications to residual architectures, Balanced ResNets are a plug-in replacement. (We even saw slightly better performance, but not statistically significant.) Note that removing hypoactivation without loss of performance is significant, as post-activation skip architectures do not have equivalent performance [2].
> > >
> > > **[2]** Kaiming He, Xiangyu Zhang, Shaoqing Ren, Jian Sun. Identity Mappings in Deep Residual Networks. ECCV (2016)  https://arxiv.org/abs/1603.05027
> > >
> > >
> > > # Proof of the Conjecture
> > > Indeed, we have made (many!) attempts at the conjecture along the lines of what you have suggested.  In the attempt you have written, the difficulty you will next hit lies with evaluating expectations of ratios of random variables of the form $\mathbb{E}\left[ \frac{X}{Y} \right]$ which is technically difficult even when both $\mathbb{E}[X]$ and $\mathbb{E}[Y]$ are known and mixed moments can be controlled. An optimistic handling of these quantities (i.e., assuming that the Taylor series can be naively truncated) leads to autoregressive formulas for the $\mathbb{E}|| \varphi_+(\hat{z}^\ell) ||$. This general autoregressive nature is confirmed by simulations (see Figure 9b in the supplemental material) and the correct order $O(1/n)$ as stated in the conjecture can be proven under the assumption that the naive Taylor expanding works. However these assumptions also turn out to predict incorrect constants for the hypoactivation! This indicates that the Taylor series used in evaluating $\mathbb{E}\left[ \frac{X}{Y} \right]$ cannot be naively truncated! Subtle technical arguments about this quantity as $n \to \infty$ are needed.
> > >
> > > One can similarly “prove” the conjecture by a completely different approach under technical approximations of random walks in $\mathbb{R}^n$ by Brownian motion $\mathbb{R}^n$. These approximations are obviously true for fixed dimension $n$, but technically very difficult in the present setting where $n \to \infty$ at the same rate as the step size of the walks $\to 0$. In other words, there are many “almost proofs” of the conjecture, all of which seem to encounter some fundamental technical obstruction which is more suited for a journal in pure math than for NeurIPS.
> > >
> > > If we were trying to minimize the role of the conjecture in our paper, we could have made one of these very technical approximations the “conjecture” and then made the currently stated Conjecture 5 a Lemma. This would have made our paper seem less conditional by obscuring the truth of the fundamental issue at play. Instead, we chose to be very transparent about the role of the conjecture in our work and invite others to build on what we have started.

---

> > > > ### Comment · Reviewer_SxtD · 2021-09-01
> > > > **response**
> > > >
> > > > Dear authors,
> > > >
> > > > Thank you for your prompt response (I imagine "delighted" expresses at least
> > > > two sentiments in this context). I find myself sympathetic to your discussion
> > > > of your work's qualitative and technical (outside of the conjecture)
> > > > contributions, and in this context feel that the decision is "borderline" in
> > > > the literal sense, so I feel that I can increase my rating and leave the
> > > > ultimate decision up to the AC and SAC (as I feel our detailed discussion has
> > > > represented the key 'hinge' issues quite clearly). However, I still feel some
> > > > need to respond on a few points, which I can elaborate on below.
> > > >
> > > > I'll use headers that match the relevant sections in your response.
> > > >
> > > > ## Path Counting
> > > >
> > > > I think this makes sense to me -- in particular, if you claim that there aren't
> > > > fundamental roadblocks to extending the calculations via your approach to the
> > > > NTK, then the position seems totally reasonable.
> > > >
> > > > ## Role of the Conjecture
> > > >
> > > > Although it's possible I'm still failing to appreciate some aspect of your
> > > > position, I don't think this represents my feelings here. My personal
> > > > perception on the conjecture is borne out of laborious analysis of the issues
> > > > of off-diagonal signal propagation in feedforward ReLU nets in the linear
> > > > regime; this activity requires the resolution of certain delicate correlation
> > > > issues, and the parallels with the hypoactivation issue (although ultimately it
> > > > seems they are distinct issues that require distinct solutions) make me feel both
> > > > that the hypoactivation conjecture should be resolvable with reasonable tools
> > > > and at less than the possible two-year timescale you mention in your response.
> > > > This also gives me a (perceived, at least) sense of the scale of the technical
> > > > task that differentiates obtaining (say) $1/2 + O(1/\sqrt{n})$ estimates for
> > > > the RHS, $1/2 + O(\log \ell / n)$ estimates for the RHS, and the conjecture's
> > > > claimed $1/2 + O(1/n)$ estimate. The first estimate should be readily
> > > > obtainable by crude truncation (but as you say, it's not good enough); the
> > > > second might be obtainable with a lazy induction strategy (but, again, it won't
> > > > suffice); and therefore one expects the third to require significant technical
> > > > insight to resolve, both on the statistical (I'm associating this with $n$) and
> > > > dynamical (I'm associating this with $d$) sides! This is why I perceive there
> > > > to be a large amount of difficulty wrapped up in the conjecture. At the same
> > > > time, I'm sympathetic to your discussion of the qualitative impact your result
> > > > can provide once the conjecture is assumed in your first message and the
> > > > "Novelty" section" (in particular, noting that it gives very different
> > > > predictions from the infinite width regime!).
> > > >
> > > > ## Interlayer Correlations / Balanced Resnets
> > > >
> > > > I continue to believe it is necessary to acknowledge prior works that involve
> > > > similar schemes to the Balanced ResNet proposal (I mentioned a work of Bai and
> > > > Lee in my review). I take your point, but also feel that the experimental
> > > > results are slightly problematic in the sense that they seem to suggest that
> > > > when a ResNet can be trained to a good level of performance, there is not much
> > > > of a need to worry about hypoactivation issues.
> > > >
> > > > ## Proof of the Conjecture
> > > >
> > > > Thanks for sharing this discussion -- sorry that I missed Figure 9. This helps
> > > > me appreciate the efforts you have taken to prove the conjecture; I feel that
> > > > it might be appropriate to include some discussions of these two proof attempts
> > > > in the revision.
> > > >
> > > > It seems inessential to discuss these matters further, but I do find myself
> > > > somewhat curious about your discussion of the Taylor approach (this is what I
> > > > had in mind). In particular, what do you mean by "incorrect constants for the
> > > > hypoactivation"? For example, I was hoping it would be possible to prove a
> > > > statement like the following: suppose $\mathrm{hypoactivation}^{\ell-1} = 1/2 +
> > > > C/n$ with $C>0$ absolute; then $\mathrm{hypoactivation}^{\ell} = 1/2 + C'/n$,
> > > > with $C' \leq C$
> > > > which we could then naively iterate. Do you mean that this approach gives
> > > > something other than $1/2$, or a wrong value (too big?) of $C'$? In addition, is
> > > > it the case that (in the context of your message) $\mathbb{E}[X]/\mathbb{E}[Y]$
> > > > is not the "right" value for the hypoactivation? I had the sense that the
> > > > denominator $Y$ would be close whp to $1$ (given $\alpha^2 + \lambda^2 = 1$),
> > > > making it feasible to pursue this kind of Taylor approach.
> > > >
> > > > I do appreciate the way you have formulated the conjecture in the present
> > > > submission (per your final remarks). The whole context seems somewhat
> > > > reminiscent of the (maybe notorious) 'gradient independence' assumption in many
> > > > old signal propagation works, which I feel reflects favorably on the argument
> > > > for acceptance you are making.

---

> > > > > ### Author Response · Authors · 2021-09-02
> > > > > **Follow-up on proof**
> > > > >
> > > > > Thank you for your quick and constructive comments. We are happy to see a mutual understanding of the paper’s contributions arising. We are committed to improving our discussion of how our contributions/techniques relate to existing work (like that of Bai and Lee) and discussing some of the many roadblocks to the proof of the conjecture we have uncovered already. We cannot help but say we were "*delighted*" to read your final paragraph:
> > > > >
> > > > > > I do appreciate the way you have formulated the conjecture in the present submission (per your final remarks). The whole context seems somewhat reminiscent of the (maybe notorious) 'gradient independence' assumption in many old signal propagation works, which I feel reflects favorably on the argument for acceptance you are making.
> > > > >
> > > > > We agree that the situation is akin to the situation around the gradient independence assumption, although we harbor some hope we've written down the right statement. While notorious, the gradient independence assumption led to technical/conceptual progress in the short/medium term  and everything was eventually sorted out by the small handful of seminal papers that nailed down the NTK precisely.
> > > > >
> > > > > We'll now respond to your final questions about our various attempts to prove the conjecture.
> > > > >
> > > > > # Question about Taylor series approach
> > > > >
> > > > > > Do you mean that this approach gives something other than 1/2, or a wrong value (too big?) of C′?
> > > > >
> > > > > Indeed, what we mean is that this approach gives the wrong value for the quantity you called $C’$ in the limit if you apply Taylor series expansion in the naive way. (Wrong in that it does not match Monte Carlo experiments.) The naive Taylor series does not give the right answer.
> > > > >
> > > > > Part of the difficulty is that the ReLU function $\varphi$ is not smooth at the origin. This means that attempting approximations under the expectation of the form $E\left[\varphi\left(X+\frac{Y}{\sqrt{n}}\right)\right]\approx E \left[\varphi(X)+\frac{Y}{\sqrt{n}}\varphi^{\prime}\left(X\right)\right]$ do not have the typical $O(n^{-1})$ error that one might naively expect because of the event that $X$ is close to $0$. (The error can be instead of size $O(n^{-1/2})$, which is too large.) One can instead use difference formulas like:
> > > > >
> > > > > $\varphi\left(X+\frac{Y}{\sqrt{n}}\right)-\varphi(X)$
> > > > > $=-\left(X+\frac{Y}{\sqrt{n}}\right) sgn(X)I\left[ \frac{|Y|}{\sqrt{n}}>|X|\right] I[XY<0]+\frac{Y}{\sqrt{n}}I\left[ X>0\right]. $
> > > > >
> > > > > and then try to bound the probability of the event $P \left(|Y|/\sqrt{n}>|X|\right)$. Unfortunately this introduces complicated terms into the expansion, which means that you need to control the probability of these events in addition to the mean and variance. In other words, to prove the hypoactivation conjecture with this approach, an induction with only the mean and variance does not seem to work; you also need to control something about the distribution that controls these events. So the approach is possible but not obvious and definitely requires a lot of care.
> > > > >
> > > > > Thanks for your consideration.

---

> > > ### Author Response · Authors · 2021-09-01
> > > **Response 1/2**
> > >
> > > Response 1/2
> > >
> > > We were delighted to receive your response to our rebuttal. We really appreciate getting the chance to engage with you.
> > >
> > > At a high level, you start with questions about the generality of our approach (is it tailored to the specific problem we solve?) and the technical "novelty" in our arguments. Your questions have sharpened our own understanding of where the novelty lies. We make our case below (see "Novelty in Our Work" section below).
> > >
> > > We appreciate your links to related work. We had read these seminal papers a while ago, and the ideas had apparently lodged themselves deep in our brains below our subconscious. We agree that there are connections to be drawn out and that readers will benefit from a discussion that draws out these connections and also differences, and gives credit where credit is rightfully due. (These connections also include some of the aspects you raise when probing the novelty of our work.)
> > >
> > > We also appreciate reading your ideas for proving Conjecture 5. Your first instincts were precisely the same as ours. We discuss the roadblocks we ran into in a section below. We want to push back gently but firmly, however, on the idea that our work should be published only when we've proven Conjecture 5:
> > >
> > > First, Conjecture 5 represents a natural encapsulation of the *technical* challenge of handling hypoactivation. The conjecture essentially states that hypoactivation is reasonably well-behaved / not too large. Note that we then take this conjecture and are able to *calculate the precise contribution of interlayer correlation* in Prop. 2 (this is elaborated on in “On the Role of the Conjecture” below).  Empirical evidence (Figure 1) shows essentially exact alignment with simulations on real networks. We did not assume the final form of our main theorem; the form was a logical consequence of the well-behavedness of the errors controlled by the conjecture. You ask about the diagonal of the NTK: we're certain that our hypoactivation assumption suffices to complete the analysis of these terms as well. Indeed, we don't see fundamental roadblocks ahead for extending our work to calculations for the NTK in the infinite depth-and-width limit.
> > >
> > > As such, Conjecture 5 will permit the community to press on with the challenge of characterizing residual networks at initialization (and perhaps beyond). Irrespective of whether the conjecture is resolved in the next two months or two years, the community can use it to push science and understanding further.
> > >
> > > But now consider the following thought experiment: what if the conjecture is indeed technically tricky and resists proof for several years. We'll be stuck, unable to publish this work or any follow on work that requires this technical result. Indeed, anyone wanting to use this conjecture will be stuck.
> > >
> > > Our Conjecture is not a weakness---it is, in retrospect, our biggest contribution. It is the magic sauce. It is likely true, given our array of experiments. Using it, the community can proceed to make progress on understanding standard residual networks. Path counting arguments don't work due to hypoactivation (to be discussed in the section “On Path Counting”). No one can proceed on this architecture without understanding hypoactivation. We've discovered a natural *minimal* assumption that allows us to prove conditional results that produce predictions that overlap with empirical measurements.
> > >
> > > If the conjecture turns out to be tricky, then whoever finally resolves it will make an important contribution to the study of deep residual networks but also reap the professional rewards of this impact. If there are any concerns that subtle aspects of the conjecture may need tweaking, it is useful to remind ourselves that we are not in pure mathematics. Machine learning races ahead of other fields for a good reason.
> > >
> > > Fundamentally, unless one believes the conjecture is straightforward, we don't see how it is reasonable to demand it be proved before the work is deemed peer reviewed and ready for consumption. Demanding all subsequent work to halt until the conjecture is resolved does not only affect our work, it affects all work that might have used this conjecture to press on.
> > >
> > > In the next message, we respond to your response in roughly the order of the comments you make. Even if a couple concerns remain, we hope that you will revise your outlook and score. No paper is perfect and, as they say, perfect is the enemy of the good. We believe our contributions merit publication.

---

> ### Author Response · Authors · 2021-08-23
> **Response to Reviewer SxtD**
>
> We would enjoy the opportunity to interact with you and answer any questions you might have after reacting to our response. Please let us know if you have any questions.

---

> ### Author Response · Authors · 2021-08-29
> **Another Nudge**
>
> We still hope to hear from you! We believe we have addressed your main concern on the technical novelty beyond feedforward networks in our initial response, and let us know if you agree or have any follow up questions and concerns.

---

> ### Author Response · Authors · 2021-08-31
> **Rebuttal discussion**
>
> Hello. We're writing again in hopes that you will respond to our rebuttal and point to specific concerns that remain so that we might have the chance to respond to them.

---

### Decision · Program_Chairs · 2021-09-27

**Decision:**

Accept (Poster)

**Comment:**

The papers studies the infinite width and depth limit of ReLU ResNets, in particular the limiting distribution of the outputs of the network, for which under certain conjectured assumptions, is claimed to be log-Gaussian. While the reviewers have mixed opinions, after an extensive discussion between authors and reviewers and among reviewers themselves, most agree on the result being interesting and novel. While there are still some questions on technicality, in particular whether some of the conjectures hold true, in view of encouraging novelty and new insights, the meta-review would recommend acceptance of the paper as a poster to the conference.